# Composing Global Optimizers to Reasoning Tasks via Algebraic Objects in Neural Nets

## Abstract

We prove rich algebraic structures of the solution space for 2-layer neural networks with quadratic activation and $L_2$ loss, trained on reasoning tasks in Abelian group (e.g., modular addition). Such a rich structure enables *analytical* construction of global optimal solutions from partial solutions that only satisfy part of the loss, despite its high nonlinearity. We coin the framework as CoGO (*Composing Global Optimizers*). Specifically, we show that the weight space over different numbers of hidden nodes of the 2-layer network is equipped with a semi-ring algebraic structure, and the loss function to be optimized consists of *monomial potentials*, which are ring homomorphisms, allowing partial solutions to be composed into global ones by ring addition and multiplication. Our experiments show that around $95\%$ of the solutions obtained by gradient descent match exactly our theoretical constructions. Although the global optimizers constructed only required a small number of hidden nodes, our analysis on gradient dynamics shows that overparameterization asymptotically decouples training dynamics and is beneficial. We further show that training dynamics favors simpler solutions under weight decay, and thus high-order global optimizers such as perfect memorization are unfavorable.

## 1 Introduction

Large Language Models (LLMs) have shown impressive results in various disciplines (OpenAI, 2024; Anthropic; Team, 2024b;a; Dubey et al., 2024; Jiang et al., 2023), while they also make surprising mistakes in basic reasoning tasks (Nezhurina et al., 2024; Berglund et al., 2023). Therefore, it remains an open problem whether it can truly do reasoning tasks. On one hand, existing works demonstrate that the models can learn efficient algorithms (e.g., dynamic programming (Ye et al., 2024) for language structure modeling, etc) and good representations (Jin & Rinard, 2024; Wijmans et al., 2023). Some reports emergent behaviors (Wei et al., 2022) when scaling up with data and model size. On the other hand, many works also show that LLMs cannot self-correct (Huang et al., 2023), and cannot generalize very well beyond the training set for simple tasks (Dziri et al., 2023; Yehudai et al., 2024; Ouellette et al., 2023), let alone complicated planning tasks (Kambhampati et al., 2024; Xie et al., 2024).

To understand how the model performs reasoning and further improve its reasoning power, people have been studying simple arithmetic reasoning problems in depth. Modular addition (Nanda et al., 2023; Zhong et al., 2024), i.e., predicting $a + b \mod d$ given $a$ and $b$, is a popular one due to its simple and intuitive structure yet surprising behaviors in learning dynamics (e.g., grokking (Power et al., 2022)) and learned representations (e.g., Fourier bases (Zhou et al., 2024)). Most works focus on various metrics to measure the behaviors and extracting interpretable circuits from trained models (Nanda et al., 2023; Varma et al., 2023; Huang et al., 2024). Analytic solutions can be constructed and/or reverse-engineered (Gromov, 2023; Zhong et al., 2024; Nanda et al., 2023) but it is not clear how to construct a systematic framework to explain and generalize the results.

In this work, we systematically analyze 2-layer neural networks with quadratic activation and $L_2$ loss on predicting the outcome of group multiplication in Abelian group $G$, which is an extension of modular addition. We find that global optimizers can be constructed *algebraically* from small partial solutions that are optimal only for parts of the loss. We achieve this by showing that (1) for the 2-layer network, there exists a **semi-ring** structure over the weights space *across different order* (i.e., number of hidden nodes or network width), with specifically defined addition and multipli-

cation (Sec. 4.1), and (2) the $L_2$ loss is a function of ***monomial potentials*** (MPs), which are ring homomorphisms (Theorem 1) that allow compositions of partial solutions into global ones using ring addition and multiplication.

As a result, our theoretical framework, named CoGO (i.e., _Composing Global Optimizers_), successfully constructs two distinct types of Fourier-based global optimizers of per-frequency order 4 (or "$2 \times 2$") and order 6 (or "$2 \times 3$"), and a global optimizer of order $d^2$ that correspond to perfect memorization. Empirically, we demonstrate that around $95\%$ of the solutions obtained from gradient descent (with weight decay) have the predicted structure and match exactly with our theoretical construction of order-4 and order-6 solutions. In addition, we also analyze the training dynamics, and show that the dynamics favors low-order global optimizers, since global optimizers algebraically connected by ring multiplication can be proven to also be topologically connected. Therefore, high-order solution like perfect memorization is unfavorable in the dynamics. When the network width goes to infinity, the dynamics of monomial potentials becomes decoupled, demystifying why over-parameterization improves the performance.

To our best knowledge, we are the first to discover such algebraic structures inside network training, apply it to analyze solutions to reasoning tasks such as modular additions, and show our theoretical constructions occur in actual gradient descent solutions.

## 2 RELATED WORKS

**Algebraic structures for maching learning**. Many works leverage symmetry and group structure in deep learning. For example, in geometric deep learning, different forms of symmetry are incorporated into network architectures (Bronstein et al., 2021). However, they do not open the black box and explore the algebraic structures of the network itself during training.

**Expressibility**. Existing works on expressibility (Li et al., 2024; Liu et al., 2022) gives explicit weight construction of neural networks weights (e.g., Transformers) for reasoning tasks like automata, which includes modular addition. However, their works do not discover algebraic structures in the weight space and loss, nor learning dynamics analysis, and it is not clear whether the constructed weights coincide with the actual solutions found by gradient descent, even in synthetic data.

**Fourier Bases in Arithmetic Tasks**. Existing works discovered that pre-trained models use Fourier bases for arithmetic operations (Zhou et al., 2024). This is true even for a simple Transformer, or even a network with one hidden layer (Morwani et al., 2023). Previous works also construct analytic Fourier solutions (Gromov, 2023) for modular addition, but with the additional assumption of infinite width, unaware of the algebraic structures we discover. Existing theoretical work (Morwani et al., 2023) also shows group-theoretical results on algebraic tasks related to finite groups, also for networks with one-hidden layers and quadratic activations. Compared to ours, they use the max-margin framework with a special regularization ($L_{2,3}$ norm) rather than $L_2$ loss, do not characterize and leverage algebraic structures in the weight space, and do not analyze the training dynamics.

## 3 DECOUPLING $L_2$ LOSS FOR REASONING TASKS OF ABELIAN GROUP

**Basic group theory**. A set $G$ forms a _group_, which means that (1) there exists an operation $\cdot$ (i.e., "multiplication"): $G \times G \mapsto G$ and it satisfies association: $(g_1 \cdot g_2) \cdot g_3 = g_1 \cdot (g_2 \cdot g_3)$. Often we write $g_1 g_2$ instead of $g_1 \cdot g_2$ for brevity. (2) there exists an identity element $e \in G$ so that $eg = ge = g$, (3) for every group element $g \in G$, there is a unique inverse $g^{-1}$ so that $gg^{-1} = g^{-1}g = e$. In some groups, the multiplication operation is commutative, i.e., $gh = hg$ for any $g, h \in G$. Such groups are called _Abelian group_. Modular addition forms a Abelian (more specifically, cyclic) group by noticing that there exists a mapping $a \mapsto e^{2\pi a \mathrm{i}/d}$ and $a + b \mod d$ is $e^{2\pi a \mathrm{i}/d} \cdot e^{2\pi b \mathrm{i}/d} = e^{2\pi (a+b)\mathrm{i}/d}$.

**Basic Ring theory**. A set $\mathcal{Z}$ forms a _ring_, if there exists two operations, addition $+$ and multiplication $*$, so that (1) $\langle \mathcal{Z}, + \rangle$ forms an Abelian group, (2) $\langle \mathcal{Z}, * \rangle$ is a monoid (i.e., a group without inverse), and (3) multiplication distributes with addition (i.e., $a * (b + c) = a * b + a * c$ and $(b + c) * a = b * a + c * a$). $\mathcal{Z}$ is called a _semi-ring_ if $\langle \mathcal{Z}, + \rangle$ is a monoid.

**Notation**. Let $\mathbb{R}$ be the real field and $\mathbb{C}$ be the complex field. For a complex vector $\boldsymbol{z}$, $\boldsymbol{z}^\top$ is its transpose, $\bar{\boldsymbol{z}}$ is its complex conjugate and $\boldsymbol{z}^*$ its conjugate transpose. For a tensor $z_{ijk}$, $z_{.jk}$ is a vector along its first dimension, $z_{i.k}$ along its second dimension, and $z_{ij.}$ along its last dimension.

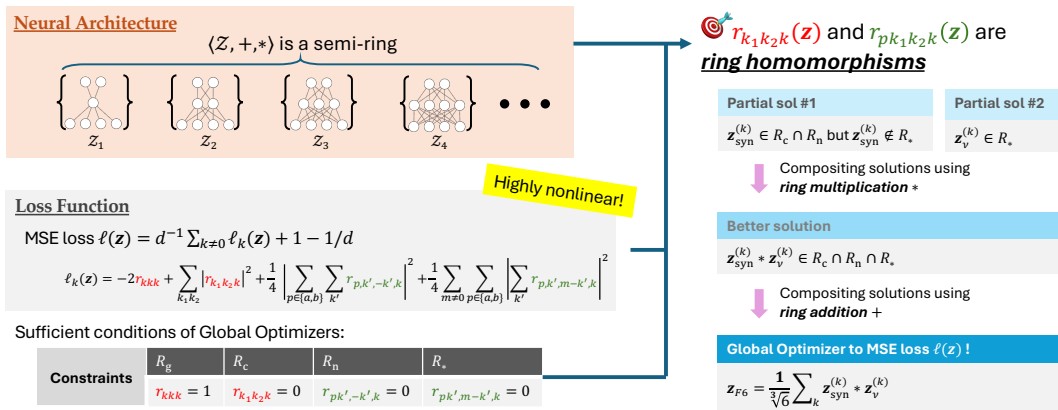

Figure 1: Overview of proposed theoretical framework CoGO. (1) The family of 2-layer neural networks, $\mathcal{Z}$, form a *semi-ring* algebraic structure (Theorem 2) with ring addition and multiplication (Def. 5). $\mathcal{Z} = \bigcup_{q \geq 0} \mathcal{Z}_q$ where $\mathcal{Z}_q$ is a collection of all weights with order-$q$ (i.e., $q$ hidden nodes). (2) For outcome prediction of Abelian group multiplication, the MSE loss $\ell(z)$ is a function of *monomial potentials* (MPs) $r_{k_1 k_2 k}(z)$ and $r_{p k_1 k_2 k}(z)$ (Theorem 1), which are ring homomorphisms (Theorem 3). (3) Thanks to the property of ring homomorphism, global optimizers to MSE loss $\ell(z)$ with quadratic activation can be constructed *algebraically* from partial solutions that only satisfy a subset of constraints (Sec. 5.1) using ring addition and multiplication, instead of running gradient descent. Examples include Fourier solution $z_{F6}$ (Corollary 2) and $z_{F4/6}$ (Corollary 4) and perfect memorization solution $z_M$ (Corollary 3). In Sec. 6, we analyze the role played of MPs in gradient dynamics, showing that the dynamics favors low-order global optimizers (Theorem 6) under weight decay regularization, and the dynamics of MPs become decoupled with infinite width (Theorem 7). In Sec. 7 we show that the gradient descent solutions match exactly with our theoretical construction.

**Problem Setup**. We consider the following 2-layer networks with $q$ hidden nodes, trained with (projected) $\ell_2$ loss on prediction of group multiplication in Abelian group $G$ with $|G| = d$:

$$\ell = \sum_i \left\| P_1^\perp \left( \frac{1}{2d} \boldsymbol{o}[i] - \boldsymbol{e}_{l[i]} \right) \right\|^2, \qquad \boldsymbol{o}[i] = \sum_j \mathbf{w}_{cj} \sigma(\mathbf{w}_{aj}^\top \boldsymbol{e}_{g_1[i]} + \mathbf{w}_{bj}^\top \boldsymbol{e}_{g_2[i]}) \tag{1}$$

**Input and Output**. The input contains the two group elements $g_1[i], g_2[i] \in G$ to be multiplied, $\boldsymbol{e}_{g_1[i]}, \boldsymbol{e}_{g_2[i]} \in \mathbb{R}^d$ are one-hot representation of $g_1[i]$ and $g_2[i]$. Here $i$ is the sample index. The target $\boldsymbol{e}_{l[i]}$ is a one-hot representation of $l[i] = g_1[i]g_2[i] \in G$, the group product of $g_1[i]$ and $g_2[i]$.

**Architectures**. In Eqn. 1, we use quadratic activation $\sigma(x) = x^2$ (Du & Lee, 2018; Allen-Zhu & Li, 2023), $P_1^\perp = I - \frac{1}{d}\mathbf{1}\mathbf{1}^\top$ is the zero-mean projection, $\mathbf{w}_{aj}, \mathbf{w}_{bj}, \mathbf{w}_{cj} \in \mathbb{R}^d$ are learnable parameters ($1 \leq j \leq q$). Note that variants of quadratic activation have been used empirically, e.g. squared ReLU and gated activations (So et al., 2021; Shazeer, 2020; Zhang et al., 2024).

We can extend our framework to *group action prediction*, in which $g_2$ may not be a group element but any object (e.g., a discrete state in reinforcement learning). See Appendix E for more details.

Let $\boldsymbol{\phi}_k = [\phi_k(g)]_{g \in G} \in \mathbb{C}^d$ be the scaled Fourier bases (or more formally, *character function* of the finite Abelian group $G$, see Appendix A). Then the weight vector $\mathcal{W} := \{\mathbf{w}_j\}$ can be written as:

$$\mathbf{w}_{aj} = \sum_{k \neq 0} z_{akj} \boldsymbol{\phi}_k, \qquad \mathbf{w}_{bj} = \sum_{k \neq 0} z_{bkj} \boldsymbol{\phi}_k, \qquad \mathbf{w}_{cj} = \sum_{k \neq 0} z_{ckj} \bar{\boldsymbol{\phi}}_k \tag{2}$$

where $\boldsymbol{z} := \{z_{pkj}\}$ are the complex coefficients, $p \in \{a, b, c\}$, $0 \leq k < d$ and $j$ runs through $q$ hidden nodes. For convenience, we define $\boldsymbol{\phi}_{-k} := \bar{\boldsymbol{\phi}}_k$ as the (complex) conjugate representation of $\boldsymbol{\phi}_k$. We exclude $\boldsymbol{\phi}_0 \equiv 1$ because the constant bias term has been filtered out by the top-down gradient from the loss function. Leveraging the property of quadratic activation functions, we can write down the loss function analytically (see Appendix A):

**Theorem 1** (Analytic form of $L_2$ loss with quadratic activation). *The objective of 2-layer MLP network with quadratic activation can be written as $\ell = d^{-1} \sum_{k \neq 0} \ell_k + (d-1)/d$, where*

$$\ell_k = -2r_{kkk} + \sum_{k_1 k_2} |r_{k_1 k_2 k}|^2 + \frac{1}{4} \left| \sum_{p \in \{a,b\}} \sum_{k'} r_{p,k',-k',k} \right|^2 + \frac{1}{4} \sum_{m \neq 0} \sum_{p \in \{a,b\}} \left| \sum_{k'} r_{p,k',m-k',k} \right|^2 \tag{3}$$

*Here $r_{k_1 k_2 k} := \sum_j z_{ak_1 j} z_{bk_2 j} z_{ckj}$ and $r_{pk_1 k_2 k} := \sum_j z_{pk_1 j} z_{pk_2 j} z_{ckj}$.*

Note that for cyclic group $G$, the frequency $k$ is a mod-$d$ integer. For general Abelian group which can be decomposed into a direct sum of cyclic groups according to Fundamental Theorem of Finite Abelian Groups (Diaconis, 1988), $k$ is a multidimensional frequency index. Since $\{\mathbf{w}_{pj}\}$ are all real, the Hermitian constraints hold, i.e. $\overline{z_{ckj}} = \overline{\boldsymbol{\phi}_k^* \mathbf{w}_{cj}} = \boldsymbol{\phi}_{-k}^* \mathbf{w}_{cj} = z_{c,-k,j}$ (and similar for $z_{akj}$ and $z_{bkj}$). Therefore, $z_{p,-k,j} = \bar{z}_{pkj}$, $r_{-k,-k,-k} = \bar{r}_{kkk}$ and $\ell$ is real and can be minimized.

Eqn. 3 contains different $r$ terms, which play an important role in determining global optimizers.

**Definition 1** (0/1-set). *Let $R := \{r\}$ be a collection of $r$ terms. The weight $\boldsymbol{z}$ is said to have 0-set $R_0$ and 1-set $R_1$ (or 0/1-sets $(R_0, R_1)$), if $r(\boldsymbol{z}) = 0$ for all $r \in R_0$ and $r(\boldsymbol{z}) = 1$ for all $r \in R_1$.*

With 0/1-sets, we can characterize rough structures of the global optimizers to the loss:

**Lemma 1** (A Sufficient Conditions of Global optimizers of Eqn. 3). *If the weight $\boldsymbol{z}$ to Eqn. 3 has 0-sets $R_c \cup R_n \cup R_*$ and 1-set $R_g$, i.e.*

$$r_{kkk}(\boldsymbol{z}) = \mathbb{I}(k \neq 0), \quad r_{k_1 k_2 k}(\boldsymbol{z}) = 0, \quad r_{pk_1 k_2 k}(\boldsymbol{z}) = 0 \tag{4}$$

*then it is a global optimizer with zero loss $\ell(\boldsymbol{z}) = 0$. Here $R_g := \{r_{kkk}, k \neq 0\}$, $R_c := \{r_{k_1 k_2 k}, k_1, k_2, k \text{ not all equal}\}$, $R_n := \{r_{p,k',-k',k}\}$ and $R_* := \{r_{p,k',m-k',k}, m \neq 0\}$.*

Lemma 1 provides *sufficient* conditions since there may exist solutions that achieve global optimum (e.g., $\sum_{k'} r_{p,k',m-k',k}(\boldsymbol{z}) = 0$ but $r_{p,k',m-k',k}(\boldsymbol{z}) \neq 0$). However, as we will see, it already leads to rich algebraic structure, and serves as a good starting point. Directly finding the global optimizers using Eqn. 4 can be a bit complicated and highly non-intuitive, due to highly nonlinear structure of Eqn. 3. However, there are nice structures we can leverage, as we will demonstrate below.

## 4 BEYOND FIXED PARAMETER SPACE: THE SEMI-RING STRUCTURE

### 4.1 THE SEMI-RING STRUCTURE OF THE SOLUTION SPACE

We define the *weight space* $\mathcal{Z}_q = \{\boldsymbol{z}\}$ to include all the weight matrices with $q$ hidden nodes ($\mathcal{Z}_0$ means an empty network), and $\mathcal{Z} = \bigcup_{q \geq 0} \mathcal{Z}_q$ be the solution space of all different number of hidden nodes. Interestingly, $\mathcal{Z}$ naturally is equipped with a *semi-ring* structure, and each term of the loss function can effective interact with such a semi-ring structure, yielding provable global optimizers, including both the Fourier solutions empirically reported in previous works (Zhou et al., 2024; Gromov, 2023), and the perfect memorization solution (Morwani et al., 2023).

To make our argument formal, we start with a few definitions.

**Definition 2** (Order of $\boldsymbol{z}$). *The order $\mathrm{ord}(\boldsymbol{z})$ of $\boldsymbol{z} \in \mathcal{Z}$ is its number of hidden nodes.*

**Definition 3** (Scalar multiplication). *$\alpha \boldsymbol{z} \in \mathcal{Z}$ is element-wise multiplication $[\alpha z_{pkj}]$ of $\boldsymbol{z} \in \mathcal{Z}$.*

**Definition 4** (Identification of $\mathcal{Z}$). *In $\mathcal{Z}$, two solutions of the same order that differ only by a permutation along hidden dimension $j$ are considered identical.*

For any two solutions $\boldsymbol{z}_1 := \{z_{pkj}^{(1)}\}$ and $\boldsymbol{z}_2 := \{z_{pkj}^{(2)}\}$, we can define their operations:

**Definition 5** (Addition and Multiplication in $\mathcal{Z}$). *Define $\boldsymbol{z} = \boldsymbol{z}_1 + \boldsymbol{z}_2$ in which $z_{pk\cdot} := \mathrm{concat}(z_{pk\cdot}^{(1)}, z_{pk\cdot}^{(2)})$ and $\boldsymbol{z} = \boldsymbol{z}_1 * \boldsymbol{z}_2$, in which $z_{pk\cdot} := z_{pk\cdot}^{(1)} \otimes z_{pk\cdot}^{(2)}$. The addition and multiplication respect Hermitian constraints and the identity element $\mathbf{1}$ is the 1-order solutions with $\{z_{pk0} = 1\}$.*

Note that the multiplication definition is one special case of Khatri–Rao product (Khatri & Rao, 1968). Although the Kronecker product and concatenation are not commutative, thanks to the identification (Def. 4), it is clear that $\boldsymbol{z}_1 + \boldsymbol{z}_2 = \boldsymbol{z}_2 + \boldsymbol{z}_1$ and $\boldsymbol{z}_1 * \boldsymbol{z}_2 = \boldsymbol{z}_2 * \boldsymbol{z}_1$ and thus both operations are commutative. Then we can show:

**Theorem 2** (Algebraic Structure of $\mathcal{Z}$). *$\langle \mathcal{Z}, +, * \rangle$ is a commutative semi-ring.*

As we will see, the semi-ring structure of $\mathcal{Z}$ paves the way to construct explicitly global optimizers.

Now let us study the structure of the loss function Eqn. 3 and how they are related to the semi-ring structure of $\mathcal{Z}$. For this, we first define the concept of *monomial potentials*:

**Definition 6** (Monomial potential (MP)). *$r(\boldsymbol{z}) := \sum_j \prod_{p,k \in \mathrm{idx}(r)} z_{pkj}$ is called monomial potential (MP), where $\mathrm{idx}(r)$ specifies the indices involved in the monomial terms.*

Following this definition, terms in the loss function (Theorem 1) are examples of MPs.

**Observation 1** (Specific MPs). *$r_{k_1 k_2 k}(\boldsymbol{z})$ and $r_{pk_1 k_2 k}(\boldsymbol{z})$ defined in Theorem 1 are MPs.*

So what is the relationship between MPs, which are functions that map a weight $\boldsymbol{z}$ to a complex scalar, and the semi-ring structure of $\mathcal{Z}$? The following theorem tells that MPs are *ring homomorphisms*, that is, these mappings respect addition and multiplication:

**Theorem 3.** *For any monomial potential $r : \mathcal{Z} \mapsto \mathbb{C}$, $r(\boldsymbol{1}) = 1$, $r(\boldsymbol{z}_1 + \boldsymbol{z}_2) = r(\boldsymbol{z}_1) + r(\boldsymbol{z}_2)$ and $r(\boldsymbol{z}_1 * \boldsymbol{z}_2) = r(\boldsymbol{z}_1)r(\boldsymbol{z}_2)$ and thus $r$ is a ring homomorphism.*

**Observation 2.** *The order function $\mathrm{ord} : \mathcal{Z} \mapsto \mathbb{N}$ is also a ring homomorphism.*

Since the loss function $\ell(\boldsymbol{z})$ depends on the weight $\boldsymbol{z}$ entirely through $r_{k_1 k_2 k}(\boldsymbol{z})$ and $r_{pk_1 k_2 k}(\boldsymbol{z})$, which are MPs, due to the property of ring homomorphism, it is possible to construct a global optimizer from partial solutions that satisfy only some of the constraints[1]:

**Lemma 2** (Composing Partial Solutions). *If $\boldsymbol{z}_1$ has 0/1-sets $(R_1^-, R_1^+)$ and $\boldsymbol{z}_2$ has 0/1-sets $(R_2^-, R_2^+)$, then (1) $\boldsymbol{z}_1 * \boldsymbol{z}_2$ has 0/1-sets $(R_1^- \cup R_2^-, R_1^+ \cap R_2^+)$. (2) $\boldsymbol{z}_1 + \boldsymbol{z}_2$ have 0/1-sets $(R_1^- \cap R_2^-, R_1^+ \cup R_2^+)$.*

Once we reach 0/1-sets $(R_c \cup R_n \cup R_*, R_g)$, we find a global optimizer. In addition, we also immediately know that there exists infinitely many global optimizers, via ring multiplication (Def. 5):

**Definition 7** (Unit). *$\boldsymbol{z}$ is called a unit if $r_{kkk}(\boldsymbol{z}) = 1$ for all $k \neq 0$.*

**Corollary 1.** *If $\boldsymbol{z}$ is a global optimizer and $\boldsymbol{y}$ is a unit, then $\boldsymbol{z} * \boldsymbol{y}$ is also a global optimizer.*

## 5   Composing Global Optimizers

### 5.1   Constructing Partial Solutions with Polynomials

While intuitively one can get global optimizers by manually crafting some partial solutions and combining, in this section, we provide a more systematic approach to compose global optimizers as follows. Since $\mathcal{Z}$ enjoys a semi-ring structure, we consider a *polynomial* in $\mathcal{Z}$ in the following form:

$$\boldsymbol{z} = \boldsymbol{u}^L + \boldsymbol{c}_1 * \boldsymbol{u}^{L-1} + \boldsymbol{c}_2 * \boldsymbol{u}^{L-2} + \ldots + \boldsymbol{c}_L \tag{5}$$

where the *generator* $\boldsymbol{u}$ and coefficients $\boldsymbol{c}_l$ are order-1 and the power operation $\boldsymbol{u}^l$ is defined by ring multiplication. The following construction of a polynomial leads to a partial solution.

**Theorem 4** (Construction of partial solutions). *Suppose $\boldsymbol{u}$ has 1-set $R_1$, $\Omega_R(\boldsymbol{u}) := \{r(\boldsymbol{u}) | r \in R\} \subseteq \mathbb{C}$ is a set of evaluations on $R$ (multiple values counted once), then if $1 \notin \Omega_R$, then the polynomial solution $\boldsymbol{\rho}_R(\boldsymbol{u}) := \prod_{s \in \Omega_R(\boldsymbol{u})}(\boldsymbol{u} + \hat{s})$ has 0/1-set $(R, R_1)$ up to a scale. Here $\hat{s}$ is any order-1 weight that satisfies $r(\hat{s}) = -s$ for any $r \in R \cup R_+$. For example, $\hat{s} = -s^{1/3}\boldsymbol{1}$.*

For convenience, we use $\boldsymbol{\rho}(\boldsymbol{u})$ to represent the *maximal* polynomial, i.e., when $R = \arg\max_{1 \notin \Omega_R(\boldsymbol{u})} |\Omega_R(\boldsymbol{u})|$ is the largest subset of MPs with $1 \notin \Omega_R(\boldsymbol{u})$. Our goal is to find low-order (partial) solutions, since gradient descent prefers low order solutions (see Theorem 6). Although there exist high-degree but low-order polynomials, e.g., $\boldsymbol{u}^9 + \boldsymbol{1}$, in general, degree $L$ and order $q$ are correlated, and we can find low-degree ones instead. To achieve that, $\boldsymbol{u}$ should be properly selected (e.g., symmetric weights) to create as many duplicate values (but not 1) in $R$ as possible.

---

[1] Mathematically, the *kernel* $\mathrm{Ker}(r) := \{\boldsymbol{z} : r(\boldsymbol{z}) = 0\}$ of a ring homomorphism $r$ is an *ideal* of the ring, and the intersection of ideals are still ideals. For brevity, we omit the formal definitions.

| | | Evaluation on MPs | | | | | | | | | Maximal | |
|---|---|---|---|---|---|---|---|---|---|---|---|---|
| | | $R_{\rm c}$ | | | $R_{\rm n}$ | | $R_*$ | | | | Maximal | |
| Symbol | $[a, b, c]$ | $\bar{a}bc$ | $a\bar{b}c$ | $ab\bar{c}$ | $\bar{a}ac$ | $\bar{b}bc$ | $aac$ | $bbc$ | $\bar{a}\bar{a}c$ | $\bar{b}\bar{b}c$ | polynomial $\boldsymbol{\rho}(\boldsymbol{u})$ | order $q$ |
| $\mathbf{1}_k$ | $[1,1,1]$ | 1 | 1 | 1 | 1 | 1 | 1 | 1 | 1 | 1 | – | – |
| $\tilde{\mathbf{1}}_k$ | $[-1,-1,1]$ | 1 | 1 | 1 | 1 | 1 | 1 | 1 | 1 | 1 | – | – |
| $\boldsymbol{u}_{\rm one}$ | $[1,-1,-1]$ | 1 | 1 | 1 | $-1$ | $-1$ | $-1$ | $-1$ | $-1$ | $-1$ | $\boldsymbol{u}+1$ | 2 |
| $\boldsymbol{u}_{\rm syn}$ | $[\omega_3,\omega_3,\omega_3]$ | $\omega_3$ | $\omega_3$ | $\omega_3$ | $\omega_3$ | $\omega_3$ | 1 | 1 | $\bar{\omega}_3$ | $\bar{\omega}_3$ | $\boldsymbol{u}^2+\boldsymbol{u}+1$ | 3 |
| $\boldsymbol{u}_{\rm 3c}$ | $[\omega_3,\bar{\omega}_3,1]$ | $\omega_3$ | $\bar{\omega}_3$ | 1 | 1 | 1 | $\bar{\omega}_3$ | $\omega_3$ | $\bar{\omega}_3$ | $\bar{\omega}_3$ | $\boldsymbol{u}^2+\boldsymbol{u}+1$ | 3 |
| $\boldsymbol{u}_{\rm 3a}$ | $[1,\omega_3,\bar{\omega}_3]$ | 1 | $\omega_3$ | $\bar{\omega}_3$ | $\bar{\omega}_3$ | $\bar{\omega}_3$ | $\bar{\omega}_3$ | $\omega_3$ | $\bar{\omega}_3$ | 1 | $\boldsymbol{u}^2+\boldsymbol{u}+1$ | 3 |
| $\boldsymbol{u}_{\rm 4c}$ | $[{\rm i},-{\rm i},1]$ | $-1$ | $-1$ | 1 | 1 | 1 | $-1$ | $-1$ | $-1$ | $-1$ | $\boldsymbol{u}+1$ | 2 |
| $\boldsymbol{u}_{\rm 4a}$ | $[1,{\rm i},-{\rm i}]$ | 1 | $-1$ | $-1$ | $-{\rm i}$ | $-{\rm i}$ | $-{\rm i}$ | ${\rm i}$ | $-{\rm i}$ | ${\rm i}$ | $\boldsymbol{u}^3+\boldsymbol{u}^2+\boldsymbol{u}+1$ | 4 |
| $\boldsymbol{u}_{\nu}$ | $[\nu,-\nu,-\bar{\nu}^2]$ | $\nu^2$ | $\nu^2$ | $\nu^4$ | $-\bar{\nu}^2$ | $-\bar{\nu}^2$ | $-1$ | $-1$ | $-\nu^4$ | $-\nu^4$ | 9-th degree | 10 |

Table 1: Exemplar order-1 single frequency generator $\boldsymbol{u}^{(k)}$ with $r_{kkk}(\boldsymbol{u}^{(k)}) = 1$. In the single-frequency case, for each MP $r$ we use "$\bar{a}bc$" to represent $r_{-k,k,k}$ and "$\bar{a}\bar{a}c$" to represent $r_{a,-k,-k,k}$, etc. We omit superscript "$(k)$" for clarity and omit conjugate columns (i.e., $\bar{a}\bar{b}c$ which is conjugate to $ab\bar{c}$). Here, $\omega_3 := e^{2\pi{\rm i}/3}$ and $\omega_4 := {\rm i}$ are the 3rd and 4th roots of unity. The constructed solutions are partial, i.e., the evaluation of some MPs yields 1 (red cell) and cannot be the root of the polynomial according to Theorem 4. Note that $\boldsymbol{u}_{\nu}$ is a general case with $\boldsymbol{u}_{\nu=1} = \boldsymbol{u}_{\rm one}$ and $\boldsymbol{u}_{\nu={\rm i}} = \boldsymbol{u}_{\rm 4c}$.

## 5.2 COMPOSING GLOBAL SOLUTIONS

We first consider the case that the generator $\boldsymbol{u}$ is only nonzero at frequency $k$ (and thus $-k$ by Hermitian constraints), but zero in other frequencies, i.e., $u_{pk'0} = 0$ for $k' \neq \pm k$. Such solutions correspond to Fourier bases in the original domain. Also, $\boldsymbol{u}$ has 1-set $R_1 = \{r_{kkk}\}$. This means that $\boldsymbol{u}$ can be characterized by three numbers $u_{ak0} = a$, $u_{bk0} = b$, and $u_{ck0} = c$ with $abc = 1$. In this case, only a subset of monomial potentials (MPs) whose indices only involve a single frequency $k$ are non-zero (e.g., $r_{k,-k,k} \in R_{\rm c}$ and $r_{b,-k,k,k} \in R_{\rm n}$), which makes our construction much easier.

Following Theorem 4, we can construct different partial solutions. Some examples are shown in Table 1, which do not reach the complete set $R_{\rm c} \cup R_{\rm n} \cup R_*$ and therefore are not global. Note that it is possible to create a generator so that all MPs are not 1 (e.g., $\boldsymbol{u}_{\rm 3c} * \boldsymbol{u}_{\rm 4a}$), but then $|\Omega_R(\boldsymbol{u})|$ will be too large, producing high-degree polynomials (e.g., $\boldsymbol{u}_{\rm 3c} * \boldsymbol{u}_{\rm 4a}$ gives a 10-th-degree polynomial).

However, utilizing these partial solutions, with Lemma 2 we can construct global optimizers:

**Corollary 2** (Order-6 global optimizers). *The following "$3 \times 2$" Fourier solutions satisfy the sufficient condition (Lemma 1) and thus are global optimizers (assuming $d$ is odd):*

$$\boldsymbol{z}_{F6} = \frac{1}{\sqrt[3]{6}} \sum_{k=1}^{(d-1)/2} \boldsymbol{z}_{\rm syn}^{(k)} * \boldsymbol{z}_{\nu}^{(k)} * \boldsymbol{y}_k \tag{6}$$

*Here $\boldsymbol{z}_{\rm syn}^{(k)} := \boldsymbol{\rho}(\boldsymbol{u}_{\rm syn}^{(k)})$ and $\boldsymbol{z}_{\nu}^{(k)} := \boldsymbol{u}_{\nu}^{(k)} + \mathbf{1}_k$ (i.e., not maximal polynomial), where $\boldsymbol{u}_{\rm syn}$ and $\boldsymbol{u}_{\nu}$ are defined in Table 1. $\boldsymbol{y}$ is an order-1 unit. As a result, $\text{ord}(\boldsymbol{z}_{F6}) = 3 \cdot 2 \cdot 1 \cdot (d-1)/2 = 3(d-1)$ and each frequency are affiliated with 6 hidden nodes (order-6).*

**Other solutions**. We may replace $\boldsymbol{u}_{\rm syn}$ and $\boldsymbol{u}_{\nu}$ with any other pairs that collectively cover all MPs. For example, $\boldsymbol{u}_{\rm syn}$ can be combined with any of $\{\boldsymbol{u}_{\rm 3c}, \boldsymbol{u}_{\rm 3a}, \boldsymbol{u}_{\rm 4a}\}$, and $\boldsymbol{u}_{\nu=\pm{\rm i}}$ can be coupled with $\boldsymbol{u}_{\rm 3a}$ or $\boldsymbol{u}_{\rm 4a}$, etc. Here we pick one with a small order. Compared to construction from Gromov (2023), ours is much more concise and does not use infinite-width approximation.

**Even** $d$. For even $d$, simply replace $(d-1)/2$ with $\lfloor (d-1)/2 \rfloor$ and add an additional order-2 term $\boldsymbol{\rho}(\boldsymbol{u}_{\rm one}) = \boldsymbol{u}_{\rm one} + \mathbf{1}$ (Tbl. 1) for the frequency $d/2$. Note that the frequency $k = d/2$ only has $r_{kkk}$, $r_{akkk}$ and $r_{bkkk}$, and all other conjugate combinations are absent. Thus $\boldsymbol{u}_{\rm one}^{(k)} + \mathbf{1}_k$ covers them all.

Fig. 2 shows a case with $d = 7$. In this case, each frequency, out of $(d-1)/2 = 3$ total number of frequencies, is associated with 6 hidden nodes. If we remove the last term in the loss that corresponds to $R_*$, then an order-3 solution suffices (i.e. $\boldsymbol{z}_{\rm syn} = \boldsymbol{\rho}(\boldsymbol{u}_{\rm syn})$).

Using polynomials, we can also construct perfect-memorization solutions. For this, we first define two generators $\boldsymbol{u}_a$ with $u_{\cdot k0}^{(\alpha)} = [\omega_d^k, 1, \bar{\omega}_d^k]\mathbb{I}(k \neq 0)$, and $\boldsymbol{u}_b$ with $u_{\cdot k0}^{(\beta)} = [1, \omega_d^k, \bar{\omega}_d^k]\mathbb{I}(k \neq 0)$. Here $\omega_d := e^{2\pi{\rm i}/d}$ is the $d$-th root of unity.

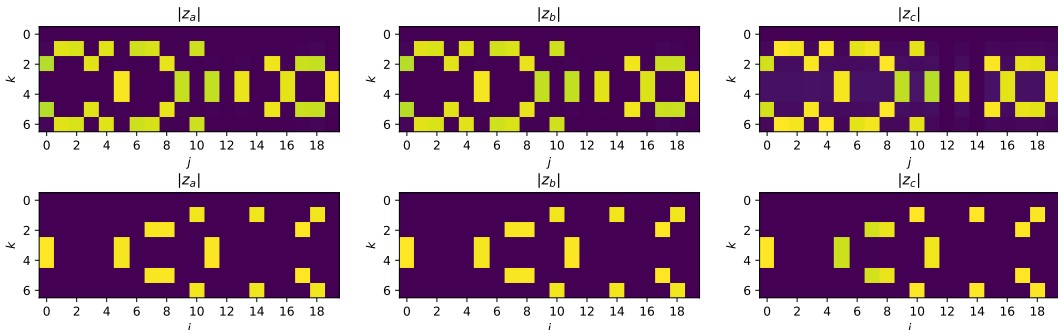

Figure 2: Solutions obtained by the Adam optimizers on $\ell_2$ loss for modular addition task with $|G| = d = 7$ and $q = 20$ hidden nodes. **Top:** For each frequency $\pm k$, there are exactly 6 hidden nodes representing such a frequency, consistent with Corollary 1. **Bottom:** Optimizing Eqn. 3 without the last term $\sum_{m \neq 0} \sum_{p \in \{a,b\}} \left| \sum_{k'} r_{p,k',m-k',k} \right|^2$ (equivalently removing the constraint $R_*$). Now each frequency has exactly 3 hidden nodes, which corresponds to the solution $\boldsymbol{z}_{\mathrm{syn}} = \boldsymbol{\rho}(\boldsymbol{u}_{\mathrm{syn}})$ in Tbl. 1.

**Corollary 3** (Perfect Memorization). *We construct two $d$-order weights $\boldsymbol{z}_a$ and $\boldsymbol{z}_b$:*

$$\boldsymbol{z}_a = \sum_{j=0}^{d-1} \boldsymbol{u}_a^j, \qquad \boldsymbol{z}_b = \sum_{j=0}^{d-1} \boldsymbol{u}_b^j \tag{7}$$

*Here $\boldsymbol{z}_a \in R_{\mathrm{c}}(k_1 \neq k) \cap R_{\mathrm{n}} \cap R_*(p = b \text{ or } m \neq k)$, $\boldsymbol{z}_b \in R_{\mathrm{c}}(k_2 \neq k) \cap R_{\mathrm{n}} \cap R_*(p = a \text{ or } m \neq k)$. Then $\boldsymbol{z}_M = d^{-2/3} \boldsymbol{z}_a * \boldsymbol{z}_b$ satisfies the sufficient condition (Lemma 1) and is the perfect memorization solution with $\mathrm{ord}(\boldsymbol{z}_M) = d^2$:*

$$z_{akj_1j_2}^{(M)} = \omega^{kj_1}/\sqrt[3]{d^2}, \qquad z_{bkj_1j_2}^{(M)} = \omega^{kj_2}/\sqrt[3]{d^2}, \qquad z_{ckj_1j_2}^{(M)} = \omega^{-k(j_1+j_2)}/\sqrt[3]{d^2} \tag{8}$$

*where each hidden node is indexed by $j = (j_1, j_2)$, $0 \leq j_1, j_2 < d$, $k \neq 0$.*

To see why this corresponds to perfect memorization, simply apply an inverse Fourier transform for each hidden node $(j_1, j_2)$, and the original weights are (zero-mean) delta function located at $j_1$, $j_2$ and $j_1 + j_2$ accordingly.

Interestingly, there also exists a lower-order solution, $2 \times 2$, that meets $R_{\mathrm{c}}$ and $R_*$ but not $R_{\mathrm{n}}$:

**Corollary 4** (Order-4 single frequency solution). *Define single frequency order-2 solution $\boldsymbol{z}_\xi$:*

$$z_{ak\cdot} = [1, \xi], \quad z_{bk\cdot} = [1, -\mathrm{i}\bar{\xi}], \quad z_{ck\cdot} = [1, \mathrm{i}] \tag{9}$$

*where $|\xi| = 1$. Then the order-4 solution $\boldsymbol{z}_{F4}^{(k)} := \boldsymbol{\rho}(\boldsymbol{u}_{\nu=\mathrm{i}}^{(k)}) * \boldsymbol{z}_\xi^{(k)}$ has 0-sets $R_{\mathrm{c}}$ and $R_*$ (but not $R_{\mathrm{n}}$).*

While $\boldsymbol{z}_{F4}^{(k)}$ itself does not satisfy the sufficient condition (Eqn. 4), it is part of a global optimizer when mixing with $\boldsymbol{z}_{F6}$:

**Corollary 5** (Mixed order-4/6 global optimizers). *With $\boldsymbol{z}_{F4}^{(k)}$, there is a global optimizer to Eqn. 3 that does not meet the sufficient condition, i.e., $\sum_{k'} r_{p,k',-k',m} = 0$ but $r_{p,k',-k',m} \neq 0$:*

$$\boldsymbol{z}_{F4/6} = \frac{1}{\sqrt[3]{6}} \hat{\boldsymbol{z}}_{F6}^{(k_0)} + \frac{1}{\sqrt[3]{4}} \sum_{k=1, k \neq k_0}^{(d-1)/2} \boldsymbol{z}_{F4}^{(k)} \tag{10}$$

*where $\hat{\boldsymbol{z}}_{F6}^{(k_0)}$ is a perturbation of $\boldsymbol{z}_{F6}^{(k_0)} := \boldsymbol{z}_{\mathrm{syn}}^{(k_0)} * \boldsymbol{z}_{\nu=1}^{(k_0)}$ by adding constant biases to its $(c, k)$ entries for $k \neq k_0$. The order is lower than $\boldsymbol{z}_{F6}$: $\mathrm{ord}(\boldsymbol{z}_{F4/6}) = 6 + 4 \cdot ((d-1)/2 - 1) = 2d < \mathrm{ord}(\boldsymbol{z}_{F6})$.*

**Remarks.** To construct $\hat{\boldsymbol{z}}_{F6}$, in addition to $\boldsymbol{z}_{\mathrm{syn}} * \boldsymbol{z}_{\nu=1}$, we could use other compositions of single frequency solutions to achieve the same effects. For example, $\boldsymbol{z}_{\mathrm{syn},\alpha\beta} * \boldsymbol{z}_{\nu=\mathrm{i}}$, where $\boldsymbol{z}_{\mathrm{syn},\alpha\beta}$ is:

$$z_{ak\cdot} = [1, \omega_3\alpha, \bar{\omega}_3\beta], \qquad z_{bk\cdot} = [1, \omega_3\bar{\alpha}, \bar{\omega}_3\bar{\beta}], \qquad z_{ck\cdot} = [1, \omega_3, \bar{\omega}_3] \tag{11}$$

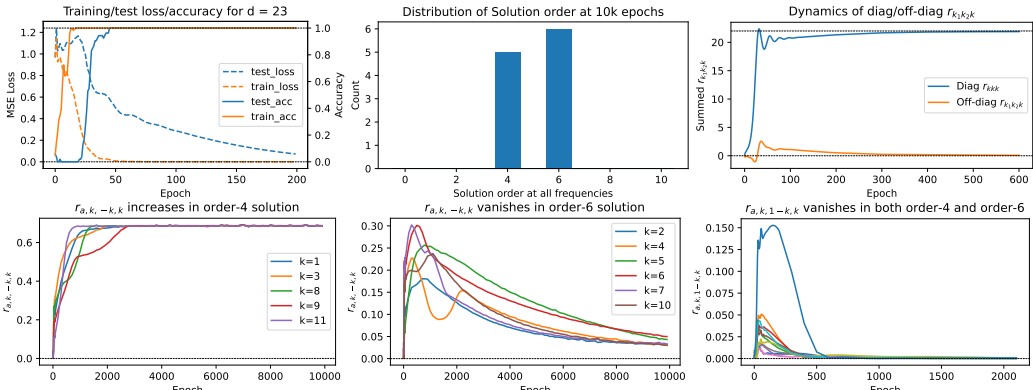

Figure 3: Dynamics of monomial potentials (MPs) over the training process for modular addition with $d = 23$ and $q = 1024$ hidden nodes. **Top Row.** *Left*: Training/test accuracy reaches 100% and loss close to 0. Test accuracy jumps after training reaches 100% (grokking). *Mid*: After 10k epochs, the distribution of solution orders are concentrated at 4 and 6 (Corollary 2 and 4). *Right*: Dynamics of $r_{k_1 k_2 k}$. Summation of diagonal $r_{kkk}$ converges towards $d - 1$ (dotted line) with ripple effects, while off-diagonal $r_{k_1 k_2 k}$ converges towards 0. **Bottom Row.** Dynamics of different MPs. Order-4 and order-6 behave differently on $r_{p,k,-k,k}$, because order-4 does not satisfy the sufficient condition (Lemma 1) but a mixture of order-4 and order-6 (i.e., $z_{F4/6}$) is still the global optimizer to the $L_2$ loss (Corollary 5).

where $|\alpha| = |\beta| = 1$. Note that $z_{\text{syn}} = \rho(u_{\text{syn}})$ is a special case of $z_{\text{syn},\alpha\beta}$ when $\alpha = \beta = 1$.

Note that multiple per frequency order-6 solutions can be inserted in this construction. Compared to all order-6 solutions $z_{F6}$, this $z_{F4/6}$ mixture solution has a lower order and is perceived in the experiments (See Fig. 6), in particular when $d$ is large (Tbl. 2), showing a strong preference of gradient descent towards lower order solutions.

## 6  GRADIENT DYNAMICS

Now we have characterized the structures of global optimizers. One natural question arises: why does the optimization procedure not converge to the perfect memorization solution $z_M$, but to the Fourier solutions $z_{F6}$ and $z_{F4/6}$? The answer is given by gradient dynamics.

Let $r = [r_{k_1 k_2 k}, r_{p k_1 k_2 k}] \in \mathbb{C}^{4d^3}$ be a vector of all MPs, and $J := \frac{\partial r}{\partial z} \frac{\partial z}{\partial \mathcal{W}}$ be the Jacobian matrix of the mapping $r = r(z(\mathcal{W}))$ in which $\mathcal{W}$ is the collection of original weights. Note that when we take derivatives with respect to $r$ and apply chain rules, we treat $r$ and its complex conjugate (e.g., $r_{kkk}$ and $r_{-k,-k,-k} = \bar{r}_{kkk}$) as independent variables. Since we run the gradient descent on $\mathcal{W}$, will such (indirect) optimization leads to a descent of $r$ towards the desired targets (Lemma 1)? This is confirmed by the following theorem:

**Theorem 5** (Dynamics of MPs). *The dynamics of MPs satisfies $\dot{r} = -JJ^* \overline{\nabla_r \ell}$, which has positive inner product with the negative gradient direction $-\overline{\nabla_r \ell}$.*

Corollary 1 shows that by ring multiplication, we could create infinitely many global optima from a base one. The following theorem answers which solution gradient dynamics picks.

**Theorem 6** (The Occam's Razer: Preference of low-order solutions). *If $z = y * z'$ and both $z$ (of order $q$) and $z'$ are global optimal solutions, then there exists a path of zero loss connecting $z$ and $z'$ in the space of $\mathcal{Z}_q$. As a result, lower-order solutions are preferred if trained with $L_2$ regularization.*

This shows that gradient dynamics with weight decay will pick a lower-order (i.e., simpler) solution, suggests that perfect memorization may not be not favorable in dynamics. The following theorem shows that the dynamics also enjoys *asymptotic freedom*:

**Theorem 7** (Infinite Width Limits at Initialization). *Considering the modified loss of Eqn. 3 with only the first two terms: $\tilde{\ell}_k := -2r_{kkk} + \sum_{k_1 k_2} |r_{k_1 k_2 k}|^2$, if the weights are i.i.d Gaussian and network width $q \to +\infty$, then $JJ^*$ converge to diagonal and the dynamics of MPs is decoupled.*

Intuitively, this means that a large enough network width ($q \to +\infty$) makes the dynamics much easier to analyze. On the other hand, the final solution may not require that large $q$. As analyzed in Corollary 2, for each frequency, to achieve global optimality, 6 hidden nodes suffice.

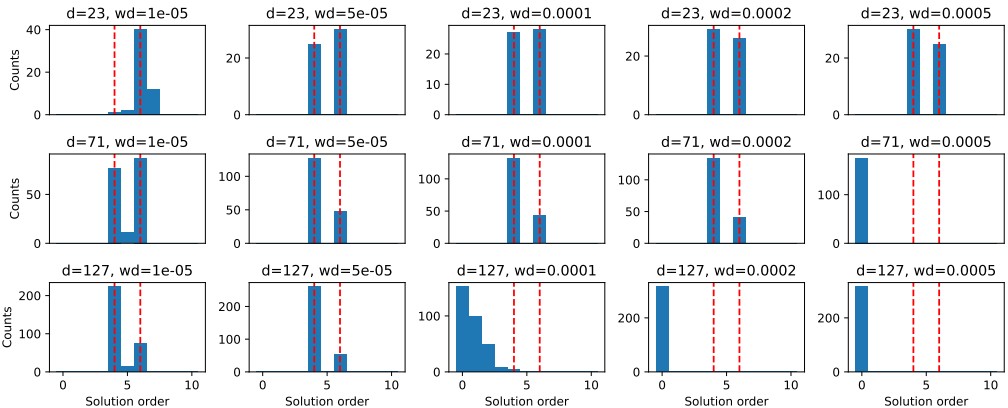

Figure 4: Solution distribution over different weight decay regularization for $q = 512$, trained with 10k epochs with Adams with learning rate 0.01 on modular addition (i.e., predicting $a+b \mod d$) with $d \in \{23, 71, 127\}$. The two red dashed lines correspond to order-4/6 solutions. The histogram is accumulated over 5 random seeds.

| $d$ | %not order-4/6 | %non-factorable order-4 | order-6 | error ($\times 10^{-2}$) order-4 | order-6 | solution distribution (%) in factorable ones $z_{\nu=i}^{(k)} * z_\xi^{(k)}$ | $z_{\nu=i}^{(k)} * z_{\mathrm{syn},\alpha\beta}^{(k)}$ | $z_\nu^{(k)} * z_{\mathrm{syn}}^{(k)}$ | others |
|---|---|---|---|---|---|---|---|---|---|
| 23 | $0.0_{\pm0.0}$ | $0.00_{\pm0.00}$ | $5.71_{\pm5.71}$ | $0.05_{\pm0.01}$ | $4.80_{\pm0.96}$ | $47.07_{\pm1.88}$ | $11.31_{\pm1.76}$ | $39.80_{\pm2.11}$ | $1.82_{\pm1.82}$ |
| 71 | $0.0_{\pm0.0}$ | $0.00_{\pm0.00}$ | $0.00_{\pm0.00}$ | $0.03_{\pm0.00}$ | $5.02_{\pm0.25}$ | $72.57_{\pm0.70}$ | $4.00_{\pm1.14}$ | $21.14_{\pm2.14}$ | $2.29_{\pm1.07}$ |
| 127 | $0.0_{\pm0.0}$ | $1.50_{\pm0.92}$ | $0.00_{\pm0.00}$ | $0.26_{\pm0.14}$ | $0.93_{\pm0.18}$ | $82.96_{\pm0.39}$ | $2.25_{\pm0.64}$ | $14.13_{\pm0.87}$ | $0.66_{\pm0.66}$ |

Table 2: Matches between order-4/6 solutions from gradient descent and those constructed by CoGO. Number of hidden nodes $q = 512$ and weight decay is $5 \times 10^{-5}$. Around 95% gradient descent solutions are factorable with very small factorization error ($\sim 0.04$ compared to solution norm on the order of 1). Furthermore, CoGO successfully predicts $\sim 98\%$ of the structure of the empirical solutions, while the remaining 2% are mostly due to insufficient training (i.e., near miss against known theoretical construction). Here $z_\xi$ is defined in Corollary 4, $z_\nu := u_\nu + 1$ is defined in Tbl. 1, and $z_{\mathrm{syn},\alpha\beta}$ is defined in Eqn. 11. The means and their standard deviations are computed over 5 seeds.

## 7 EXPERIMENTS

**Setup**. We train the 2-layer MLP on the modular addition task, which is a special case of outcome prediction of Abelian group multiplication. We use Adam optimizer with learning rate 0.01, MSE loss, and train for 10000 epochs with weight decays. We tested on $|G| = d \in \{23, 71, 127\}$. All data are generated synthetically and training/test split is 90%/10%.

**Solution Distributions**. As shown in Fig. 3, we see order-4 and order-6 solutions in each frequency emerging from well-trained networks on $d = 23$. The mixed solution $z_{F4/6}$ can be clearly observed in a small-scale example (Fig. 6). This is also true for larger $d$ (Fig. 4). Although the model is trained with heavily over-parameterized networks, the final solution order remains constant, which is consistent with Corollary 1. Large weight decay shifts the distribution to the left (i.e., low-order solutions) until model collapses (i.e., all weights become zero), consistent with our Theorem 6 that demonstrates that gradient descent with weight decay favors low-order solutions. Similar conclusions follow for fewer and more overparameterization (Appendix H).

**Exact match between theoretical construction and empirical solutions**. A follow-up question arises: *do the empirical solutions match exactly with our constructions?* After all, distribution of solution order is a rough metric. For this, we identify all solutions obtained by gradient descent at each frequency, factorize them and compare with theoretical construction up to conjugation/normalization. To find such a factorization, we use exhaustive search (Appendix H).

The answer is yes. Tbl. 2 shows that around 95% of order-4 and order-6 solutions from gradient descent can be factorized into $2 \times 2$ and $2 \times 3$ and each component matches our theoretical construction in Corollary 2 and 4, with minor variations. Furthermore, when $d$ is large, most of the solutions become order-4, which is consistent with our analysis for mixed solution $z_{F4/6}$ (Corollary 5) that one order-6 solution in the form of $z_{\nu=i} * z_{\mathrm{syn},\alpha\beta}$ suffices to achieve a global optimizer, with all other frequencies taking order-4s. In fact, for $d = 127$, the number of order-6 solution taking the form of $z_{\nu=i} * z_{\mathrm{syn},\alpha\beta}$ is $(d-1)/2 \cdot 2.25\% \approx 1.26$, coinciding with the theoretical results.

**Implicit Bias of gradient descent**. Our construction gives other possible solutions (e.g., $z_{3c} * z_{\mathrm{syn}}$) which are never observed in the gradient solutions. Even for the observed solutions, e.g. $z_\nu * z_{\mathrm{syn}}$,

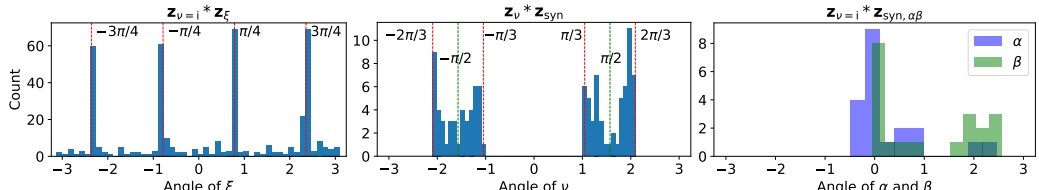

Figure 5: Distribution of free parameters ($\xi$, $\nu$, $\alpha$ and $\beta$, all with magnitude 1) in three kinds of gradient descent solutions identified by CoGO. While any value of these parameters makes a global optimizer, gradient descent dynamics has a particular preference in picking them during optimization.

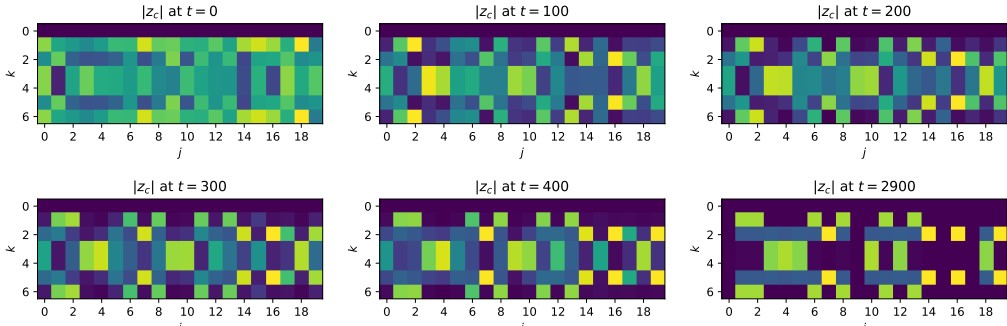

Figure 6: The convergence path of $z_{c..}$ when training modular addition using Adam optimizer (learning rate 0.05, weight decay 0.005). The final solution contains 2 order-6 ($z_{F6}^{(k)}$) and 1 order-4 ($z_{F4}^{(k)}$) solutions. Note that for $z_{c..}$, unlike Fig. 2, each order-6 solution contains a constant bias term to cancel out the artifacts of order-4 solution (Corollary 5). For each hidden node $j$, once a dominant frequency emerges, others fade away.

the distribution of free parameters is highly non-uniform (Fig. 5), showing a strong preference of certain choices. These suggest strong implicit bias in optimization, which we leave for future work.

## 8 CONCLUSION AND FUTURE WORK

In this work, we propose CoGO (*Composing Global Optimizers*), a theoretical framework that models the algebraic structure of global optimizers when training a 2-layer network on reasoning tasks of Abelian group with $L_2$ loss. We find that the global optimizers can be algebraically composed by partial solutions that only fit parts of the loss, using ring operations defined in the weight space of the 2-layer neural networks across different network widths. Under CoGO, we also analyze the training dynamics, show the benefit of over-parameterization, and the inductive bias towards simpler solutions due to topological connectivity between algebraically linked high-order (i.e., involving more hidden nodes) and low-order global optimizers. Finally, we show that the gradient descent solutions exactly match what constructed solutions (e.g. $z_{F4/6}$ and $z_{F6}$, see Corollary 5 and Corollary 2).

**Develop novel training algorithms**. Instead of applying (stochastic) gradient descent to overparameterized networks, CoGO suggests a completely different path: decompose the loss, find the MPs, construct low-order solutions and combine them to achieve the final solutions on the fly using algebraic operations. Such an approach may be more efficient and scalable than gradient descent, due to its factorable nature. Also, our framework works for losses depending on monomial potentials ($L_2$ loss is just one example), which opens a new dimension for loss design.

**Putting different widths into the same framework**. Many existing theoretical works study properties of networks with fixed width. However, CoGO demonstrates that nice mathematical structures emerge when putting networks of different widths together, an interesting direction to consider.

**Grokking**. When learning modular addition, there exists a phase transition from *memorization* to *generalization* during training, known as *grokking* (Varma et al., 2023; Power et al., 2022), long after the training performance becomes (almost) perfect. Our work may be expanded to a nonuniformly distributed training set to study the dynamics of representation learning on grokking.

**Extending to other activations.** For other activation than quadratic (e.g., SiLU) with $\sigma(0) = 0$, with a Taylor expansion, the same framework may still apply (with higher rank MPs).

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

## A  DECOUPLING $L_2$ LOSS (PROOF)

We use the *character function* $\phi : G \to \mathbb{C}$, which maps a group element $g$ into a complex number.

**Lemma 3.** *For finite Abelian group, the character function $\phi$ has the following properties Fulton & Harris (2013); Steinberg (2009):*

- *It is a 1-dimensional (irreducible) representation of the group $G$, i.e., $|\phi(g)| = 1$ for $g \in G$ and for any $g_1, g_2 \in G$, $\phi(g_1 g_2) = \phi(g_1)\phi(g_2)$.*

- *There exists $d$ character functions $\{\phi_k\}$ that satisfy the orthonormal condition $\frac{1}{d} \sum_{g \in G} \phi_k(g)\overline{\phi_{k'}}(g) = \mathbb{I}(k = k')$. Here $\overline{\phi}$ is the complex conjugate of $\phi$ and is also a character function.*

- *The set of character functions $\{\phi_k\}$ forms a* character group $\hat{G}$ *under pairwise multiplication: $\phi_{k_1 + k_2} = \phi_{k_1} \circ \phi_{k_2}$.*

Note that the *frequency* $k$ goes from $0$ to $d-1$, where $\phi_0 \equiv 1$ is the trivial representation (i.e., all $g \in G$ maps to 1). According to the Fundamental Theorem of Finite Abelian Groups, each finite Abelian group can be decomposed into a direct sum of cyclic groups, and the character function of each cyclic group is exactly (scaled) Fourier bases. Therefore, in Abelian group, $k$ is a multi-dimensional frequency index. Conrad (2010) shows that $\hat{G} \cong G$ (Theorem 3.13) so each character function $\phi \in \hat{G}$ can also be indexed by $g$ itself. Right now we keep the index $k$.

For convenience, we define $\phi_{-k} := \overline{\phi}_k$ as the (complex) conjugate representation of $\phi_k$.

Let $\boldsymbol{\phi}_k = [\phi_k(g)]_{g \in G} \in \mathbb{C}^d$ be the vector that contains the value of the character function $\phi_k$ over $G$. Then $\{\boldsymbol{\phi}_k\}$ form an orthogonal base in $\mathbb{C}^d$ and we can represent the weight vector $\mathbf{w}_{pj}$ as the following, where $p \in \{a, b, c\}$:

$$\mathbf{w}_{aj} = \sum_{k \neq 0} z_{akj}\boldsymbol{\phi}_k, \qquad \mathbf{w}_{bj} = \sum_{k \neq 0} z_{bkj}\boldsymbol{\phi}_k, \qquad \mathbf{w}_{cj} = \sum_{k \neq 0} z_{ckj}\bar{\boldsymbol{\phi}}_k \tag{12}$$

where $\boldsymbol{z} := \{z_{pkj}\}$ are the complex coefficients. Here $p \in \{a, b, c\}$, $0 \leq k < d$ and $j$ runs through hidden nodes.

**Theorem 1** (Analytic form of $L_2$ loss with quadratic activation). *The objective of 2-layer MLP network with quadratic activation can be written as $\ell = d^{-1} \sum_{k \neq 0} \ell_k + (d-1)/d$, where*

$$\ell_k = -2r_{kkk} + \sum_{k_1 k_2} |r_{k_1 k_2 k}|^2 + \frac{1}{4}\Big| \sum_{p \in \{a,b\}} \sum_{k'} r_{p,k',-k',k} \Big|^2 + \frac{1}{4} \sum_{m \neq 0} \sum_{p \in \{a,b\}} \Big| \sum_{k'} r_{p,k',m-k',k} \Big|^2 \tag{3}$$

*Here $r_{k_1 k_2 k} := \sum_j z_{ak_1 j} z_{bk_2 j} z_{ckj}$ and $r_{pk_1 k_2 k} := \sum_j z_{pk_1 j} z_{pk_2 j} z_{ckj}$.*

*Proof.* Note that the objective $\ell$ can be written down as

$$\ell = \mathbb{E}_{g_1, g_2}\left[\|P_1^{\perp}(\boldsymbol{o}(g_1, g_2)/2d - \boldsymbol{e}_{g_1 g_2})\|^2\right] \tag{13}$$

$$= \mathbb{E}_{g_1, g_2}\left[\boldsymbol{o}^{\top} P_1^{\perp} \boldsymbol{o}/4d^2 - \boldsymbol{o}^{\top} P_1^{\perp} \boldsymbol{e}_{g_1 g_2}/d + \boldsymbol{e}_{g_1 g_2}^{\top} P_1^{\perp} \boldsymbol{e}_{g_1 g_2}\right] \tag{14}$$

For notation brevity, let $z_{akj} := a_{kj}$, $z_{bkj} := b_{kj}$ and $z_{ckj} := c_{kj}$. For $\mathbb{E}\left[\boldsymbol{o}^{\top} P_1^{\perp} \boldsymbol{e}_{g_1 g_2}\right]$, since

$$\boldsymbol{e}_{g_1 g_2}^{\top} P_1^{\perp} \boldsymbol{o} = \sum_j \boldsymbol{e}_{g_1 g_2}^{\top} P_1^{\perp} \mathbf{w}_{cj} \sigma(\mathbf{w}_{aj}^{\top} \boldsymbol{e}_{g_1} + \mathbf{w}_{bj}^{\top} \boldsymbol{e}_{g_2}) \tag{15}$$

$$= \sum_j \left( \sum_{k' \neq 0} c_{k'j} \bar{\phi}_{k'}(g_1 g_2) \right) \left( \mathbf{w}_{aj}^{\top} \boldsymbol{e}_{g_1} + \mathbf{w}_{bj}^{\top} \boldsymbol{e}_{g_2} \right)^2 \tag{16}$$

$$= \sum_j \left( \sum_{k' \neq 0} c_{k'j} \bar{\phi}_{k'}(g_1 g_2) \right) \left( \sum_k \sum_{p \in \{a,b\}} z_{pkj} \phi_k(g_p) \right)^2 \tag{17}$$

Therefore, leveraging the fact that $\bar{\phi}_{k'}(g_1 g_2) = \bar{\phi}_{k'}(g_1)\bar{\phi}_{k'}(g_2)$, we have:

$$\mathbb{E}_{g_1,g_2}\left[e_{g_1 g_2}^\top P_1^\perp o\right] = \sum_{k_1,k_2,k'\neq 0,p_1,p_2,j} c_{k'j} z_{p_1 k_1 j} z_{p_2 k_2 j}\mathbb{E}_{g_1,g_2}\left[\bar{\phi}_{k'}(g_1)\bar{\phi}_{k'}(g_2)\phi_{k_1}(g_{p_1})\phi_{k_2}(g_{p_2})\right] \tag{18}$$

Since $\mathbb{E}_g\left[\phi_k(g)\bar{\phi}_{k'}(g)\right] = \mathbb{I}(k = k')$, there are only a few cases that the summand is nonzero:

- $p_1 = a$, $p_2 = b$, $k' = k_1 = k_2 \neq 0$.

- $p_1 = b$, $p_2 = a$, $k' = k_1 = k_2 \neq 0$.

In both cases, the summation reduces to $\sum_{k\neq 0,j} c_{kj} z_{akj} z_{bkj} = \sum_{k\neq 0,j} c_{kj} a_{kj} b_{kj}$. Let $r_{k_1 k_2 k'} := \sum_j a_{k_1 j} b_{k_2 j} c_{k'j}$, then we have

$$\mathbb{E}_{g_1,g_2}\left[o^\top(g_1,g_2) P_1^\perp e_{g_1 g_2}\right] = 2\sum_{k\neq 0,j} a_{kj} b_{kj} c_{kj} = 2\sum_{k\neq 0} r_{kkk} \tag{19}$$

For $\mathbb{E}\left[o^\top P_1^\perp o\right]$, we have:

$$o^\top P_1^\perp o = \sum_{j,j'} \mathbf{w}_{cj}^\top P_1^\perp \mathbf{w}_{cj'}\sigma(\mathbf{w}_{aj}^\top e_{g_1} + \mathbf{w}_{bj}^\top e_{g_2})\sigma(\mathbf{w}_{aj'}^\top e_{g_1} + \mathbf{w}_{bj'}^\top e_{g_2}) \tag{20}$$

here

$$\mathbf{w}_{cj}^\top P_1^\perp \mathbf{w}_{cj'} = \left(\sum_{k'\neq 0} c_{k'j}\bar{\phi}_{k'}\right)^\top\left(\sum_{k''\neq 0} \bar{c}_{k''j'}\phi_{k''}\right) = d\sum_{k'\neq 0} c_{k'j}\bar{c}_{k'j'} \tag{21}$$

due to the fact that $\bar{\phi}_k^\top \phi_{k'} = \sum_g \bar{\phi}_k(g)\phi_{k'}(g) = d\mathbb{I}(k = k')$.

Then the key part is to compute the following terms:

$$\mathbb{E}_{g_1,g_2}\left[z_{p_1 k_1 j_1} z_{p_2 k_2 j_1} z_{p_3 k_3 j_2} z_{p_4 k_4 j_2} c_{k'j_1}\bar{c}_{k'j_2}\phi_{k_1}(g_{p_1})\phi_{k_2}(g_{p_2})\phi_{k_3}(g_{p_3})\phi_{k_4}(g_{p_3})\right] \tag{22}$$

summing over $\{p_1, p_2, p_3, p_4, k_1, k_2, k_3, k_4, k' \neq 0, j_1, j_2\}$. Note that since each $p \in \{a, b\}$, there are $2^4 = 16$ choices of $(p_1, p_2, p_3, p_4)$. For notation brevity, we use $(1, 3)$ to represent the subset of $p$ that takes the value of $a$ (e.g., $(1, 3)$ means that $p_1 = p_3 = a$ and $p_2 = p_4 = b$). It is clear that for odd assignments such as $(1, 2, 3)$, since $z_{p0j} = 0$, the summation is zero. Then, we only discuss the even cases as follows:

**Case 1:** $(1, 3)$, $(2, 4)$, $(1, 4)$, $(2, 3)$. The 4 cases are identical so we only need to analyze one. We take $(1, 3)$ as an example. For $(1, 3)$, $p_1 = p_3 = a$, $p_2 = p_4 = b$ and the only nonzero terms is when $k_1 + k_3 = 0 \mod d$, $k_2 + k_4 = 0 \mod d$, since $\mathbb{E}_{g_1}\left[\phi_{k_1}(g_1)\phi_{k_3}(g_1)\right] = \mathbb{I}(k_1 + k_3 = 0 \mod d)$ (and similar in other cases). Then Eqn. 22 becomes:

$$\sum_{k_1,k_2,k'\neq 0}\sum_{j_1 j_2} z_{ak_1 j_1} z_{bk_2 j_1} z_{a,-k_1,j_2} z_{b,-k_2,j_2} c_{k'j_1}\bar{c}_{k'j_2} \tag{23}$$

$$= \sum_{k_1,k_2,k'\neq 0}\sum_{j_1} z_{ak_1 j_1} z_{bk_2 j_1} c_{k'j_1}\overline{\sum_{j_2} z_{ak_1 j_2} z_{bk_2 j_2} c_{k'j_2}} \tag{24}$$

$$= \sum_{k_1,k_2,k'\neq 0}\sum_{j_1} a_{k_1 j_1} b_{k_2 j_1} c_{k'j_1}\overline{\sum_{j_2} a_{k_1 j_2} b_{k_2 j_2} c_{k'j_2}} \tag{25}$$

$$= \sum_{k_1,k_2,k'\neq 0} r_{k_1 k_2 k'}\overline{r_{k_1 k_2 k'}} = \sum_{k_1,k_2,k'\neq 0} |r_{k_1 k_2 k'}|^2 \tag{26}$$

Since there are 4 such cases, we have:

$$\epsilon_1 = 4\sum_{k'\neq 0}\sum_{k_1 k_2} |r_{k_1 k_2 k'}|^2 \tag{27}$$

**Case 2:** $(1,2)$ **and** $(3,4)$. The two cases are identical. Take $(1,2)$ as an example. In this case, $p_1 = p_2 = a$ and $p_3 = p_4 = b$. The only non-zero terms are when $k_1 + k_2 = 0$, $k_3 + k_4 = 0$. Then Eqn. 22 becomes:

$$\sum_{k_1,k_3,k'\neq 0} \sum_{j_1 j_2} z_{ak_1j_1}\bar{z}_{ak_1j_1} z_{bk_3j_2}\bar{z}_{bk_3j_2} c_{k'j_1}\bar{c}_{k'j_2} \tag{28}$$

$$= \sum_{k_1,k_3,k'\neq 0} \sum_{j_1} |a_{k_1j_1}|^2 c_{k'j_1} \sum_{j_2} |b_{k_3j_2}|^2 \bar{c}_{k'j_2} \tag{29}$$

$$= \sum_{k'\neq 0} \left[ \sum_{j_1} \left( \sum_{k_1} |a_{k_1j_1}|^2 \right) c_{k'j_1} \right] \left[ \sum_{j_2} \left( \sum_{k_3} |b_{k_3j_2}|^2 \right) \bar{c}_{k'j_2} \right] \tag{30}$$

Let $r^{\circledast}_{amk'} := \sum_j \left( \sum_{k_1+k_2=m} a_{k_1j}a_{k_2j} \right) c_{k'j}$ (similar for $r^{\circledast}_{bmk'}$), then the above becomes $\sum_{k'\neq 0} r^{\circledast}_{a0k'}\bar{r}^{\circledast}_{b0k'}$.

Similarly, for $(3,4)$, the above equation becomes $\sum_{k'\neq 0} \bar{r}^{\circledast}_{a0k'}r^{\circledast}_{b0k'}$. Therefore, we have:

$$\epsilon_2 = \sum_{k'\neq 0} r^{\circledast}_{a0k'}\bar{r}^{\circledast}_{b0k'} + \bar{r}^{\circledast}_{a0k'}r^{\circledast}_{b0k'} \tag{31}$$

Note that this term can be negative. However, we will see that when it is combined with the following terms, all terms will be non-negative.

**Case 3:** $(1,2,3,4)$ **and** $()$. In this case we have:

$$\sum_{k'\neq 0} \sum_{j_1 j_2} \sum_{p\in\{a,b\}} \sum_{k_1+k_2+k_3+k_4=0} z_{pk_1j_1} z_{pk_2j_1} z_{pk_3j_2} z_{pk_4j_2} c_{k'j_1}\bar{c}_{k'j_2} \tag{32}$$

$$= \sum_{k'\neq 0} \sum_{j_1 j_2} \sum_{p\in\{a,b\}} \sum_{k_1+k_2=k_3+k_4} z_{pk_1j_1} z_{pk_2j_1} \bar{z}_{pk_3j_2} \bar{z}_{pk_4j_2} c_{k'j_1}\bar{c}_{k'j_2} \tag{33}$$

$$= \sum_{k'\neq 0} \sum_m \sum_{p\in\{a,b\}} \sum_{j_1 j_2} \sum_{k_1+k_2=m} \sum_{k_3+k_4=m} z_{pk_1j_1} z_{pk_2j_1} \bar{z}_{pk_3j_2} \bar{z}_{pk_4j_2} c_{k'j_1}\bar{c}_{k'j_2} \tag{34}$$

$$= \sum_{k'\neq 0} \sum_m \sum_{p\in\{a,b\}} \left[ \sum_{j_1} \left( \sum_{k_1+k_2=m} z_{pk_1j_1} z_{pk_2j_1} \right) c_{k'j_1} \right] \left[ \sum_{j_2} \left( \sum_{k_3+k_4=m} \overline{z_{pk_3j_2} z_{pk_4j_2}} \right) \bar{c}_{k'j_2} \right]$$

$$= \sum_{k'\neq 0} \sum_m |r^{\circledast}_{amk'}|^2 + |r^{\circledast}_{bmk'}|^2 \tag{35}$$

In particular, when $m = 0$, we have $\sum_{k'\neq 0} |r^{\circledast}_{a0k'}|^2 + |r^{\circledast}_{b0k'}|^2$. Therefore, we have

$$\epsilon_2 + \epsilon_{3,m=0} = \sum_{k'\neq 0} |r^{\circledast}_{a0k'} + r^{\circledast}_{b0k'}|^2 \tag{36}$$

Finally, putting them together, we have:

$$\mathbb{E}\left[o^\top P_1^\perp o\right] = d(\epsilon_1 + \epsilon_2 + \epsilon_3) = d(\epsilon_1 + (\epsilon_2 + \epsilon_{3,m=0}) + \epsilon_{3,m\neq 0}) \tag{37}$$

$$= d\sum_{k'\neq 0} \left( 4\sum_{k_1 k_2} |r_{k_1 k_2 k'}|^2 + |r^{\circledast}_{a0k'} + r^{\circledast}_{b0k'}|^2 + \sum_{m\neq 0} |r^{\circledast}_{amk'}|^2 + |r^{\circledast}_{bmk'}|^2 \right)$$

$$\geq 0 \tag{38}$$

Putting them together, we arrived at the conclusion. $\qquad\square$

**Lemma 1** (A Sufficient Conditions of Global optimizers of Eqn. 3). *If the weight $z$ to Eqn. 3 has 0-sets $R_c \cup R_n \cup R_*$ and 1-set $R_g$, i.e.*

$$r_{kkk}(z) = \mathbb{I}(k \neq 0), \quad r_{k_1 k_2 k}(z) = 0, \quad r_{pk_1 k_2 k}(z) = 0 \tag{4}$$

*then it is a global optimizer with zero loss $\ell(z) = 0$. Here $R_g := \{r_{kkk}, k \neq 0\}$, $R_c := \{r_{k_1 k_2 k}, k_1, k_2, k \text{ not all equal}\}$, $R_n := \{r_{p,k',-k',k}\}$ and $R_* := \{r_{p,k',m-k',k}, m \neq 0\}$.*

*Proof.* Note that $2\sum_k r_{kkk} - \sum_k |r_{kkk}|^2$ has a minimizer $r_{kkk} = 1$. Therefore, the best loss value any assignment of weights is able to achieve is the following:

$$r_{k_1 k_2 k'} = \sum_j a_{k_1 j} b_{k_2 j} c_{k' j} = \mathbb{I}(k_1 = k_2 = k') \qquad k' \neq 0 \qquad (39)$$

$$r_{a0k'}^{\circledast} + r_{b0k'}^{\circledast} := \sum_j \left( \sum_k |a_{kj}|^2 + |b_{kj}|^2 \right) c_{k'j} = 0 \qquad k' \neq 0 \qquad (40)$$

$$r_{amk'}^{\circledast} := \sum_j \left( \sum_{k_1 + k_2 = m} a_{k_1 j} a_{k_2 j} \right) c_{k'j} = 0 \qquad k' \neq 0, m \neq 0 \qquad (41)$$

$$r_{bmk'}^{\circledast} := \sum_j \left( \sum_{k_1 + k_2 = m} b_{k_1 j} b_{k_2 j} \right) c_{k'j} = 0 \qquad k' \neq 0, m \neq 0 \qquad (42)$$

Therefore the sufficient conditions (Eqn. 4) will make all above come true. $\qquad\square$

## B  SEMI-RING STRUCTURE OF $\mathcal{Z}$ (PROOF)

**Theorem 2** (Algebraic Structure of $\mathcal{Z}$). $\langle \mathcal{Z}, +, * \rangle$ *is a commutative semi-ring.*

*Proof.* Straightforward from the definition of addition and multiplication (Def. 5) and identification of hidden nodes under permutation (Def. 4). Note that ring addition (i.e., concatenation) does not have inverse and thus it is a semi-ring. $\qquad\square$

**Theorem 3.** *For any monomial potential* $r : \mathcal{Z} \mapsto \mathbb{C}$, $r(\mathbf{1}) = 1$, $r(\mathbf{z}_1 + \mathbf{z}_2) = r(\mathbf{z}_1) + r(\mathbf{z}_2)$ *and* $r(\mathbf{z}_1 * \mathbf{z}_2) = r(\mathbf{z}_1)r(\mathbf{z}_2)$ *and thus* $r$ *is a ring homomorphism.*

*Proof.* Let $r(\mathbf{z}) = \sum_j \prod_{(p,k)\in\text{idx}(r)} z_{pkj}$. Since the ring identity $\mathbf{1}$ is order-1 and all $z_{pkj} = 1$, it is obvious that $r(\mathbf{1}) = 1$.

Let $\text{supp}(\mathbf{z}_1)$ be the subset of the hidden nodes that corresponds to $\mathbf{z}_1$ in the concatenated solution $\mathbf{z}_1 + \mathbf{z}_2$, similar for $\text{supp}(\mathbf{z}_2)$. Note that

$$r(\mathbf{z}_1 + \mathbf{z}_2) = \sum_{j\in\text{supp}(\mathbf{z}_1)} \prod_{(p,k)\in\text{idx}(r)} z_{pkj}^{(1)} + \sum_{j\in\text{supp}(\mathbf{z}_2)} \prod_{(p,k)\in\text{idx}(r)} z_{pkj}^{(2)} = r(\mathbf{z}_1) + r(\mathbf{z}_2) \qquad (43)$$

On the other hand, we have

$$r(\mathbf{z}_1 * \mathbf{z}_2) = \sum_{j_1 j_2} \prod_{(p,k)\in\text{idx}(r)} \left( z_{pkj_1}^{(1)} z_{pkj_2}^{(2)} \right) \qquad (44)$$

$$= \sum_{j_1 j_2} \left( \prod_{(p,k)\in\text{idx}(r)} z_{pkj_1}^{(1)} \right) \left( \prod_{(p,k)\in\text{idx}(r)} z_{pkj_2}^{(2)} \right) \qquad (45)$$

$$= \left( \sum_{j_1} \prod_{(p,k)\in\text{idx}(r)} z_{pkj_1}^{(1)} \right) \left( \sum_{j_2} \prod_{(p,k)\in\text{idx}(r)} z_{pkj_2}^{(1)} \right) \qquad (46)$$

$$= r(\mathbf{z}_1)r(\mathbf{z}_2) \qquad (47)$$

$\qquad\square$

**Corollary 1.** *If $\mathbf{z}$ is a global optimizer and $\mathbf{y}$ is a unit, then $\mathbf{z} * \mathbf{y}$ is also a global optimizer.*

*Proof.* Straightforward by leveraging the property of ring homomorphism. E.g.,

$$r_{kkk}(\mathbf{z} * \mathbf{y}) = r_{kkk}(\mathbf{z}) r_{kkk}(\mathbf{y}) = r_{kkk}(\mathbf{z}) \qquad (48)$$

and the proof is complete. $\qquad\square$

# C  SOLUTION CONSTRUCTION (PROOF)

## C.1  CONSTRUCTION OF PARTIAL SOLUTIONS

**Theorem 4** (Construction of partial solutions). *Suppose $\boldsymbol{u}$ has 1-set $R_1$, $\Omega_R(\boldsymbol{u}) := \{r(\boldsymbol{u})|r \in R\} \subseteq \mathbb{C}$ is a set of evaluations on $R$ (multiple values counted once), then if $1 \notin \Omega_R$, then the polynomial solution $\boldsymbol{\rho}_R(\boldsymbol{u}) := \prod_{s \in \Omega_R(\boldsymbol{u})}(\boldsymbol{u} + \hat{\boldsymbol{s}})$ has 0/1-set $(R, R_1)$ up to a scale. Here $\hat{\boldsymbol{s}}$ is any order-1 weight that satisfies $r(\hat{\boldsymbol{s}}) = -s$ for any $r \in R \cup R_+$. For example, $\hat{\boldsymbol{s}} = -s^{1/3}\mathbf{1}$.*

*Proof.* By definition, for any $r \in R$ we have:

$$r(\boldsymbol{z}(\boldsymbol{u})) = \prod_{s \in \Omega_R(\boldsymbol{u})} (r(\boldsymbol{u}) + r(\hat{\boldsymbol{s}})) = \prod_{s \in \Omega_R(\boldsymbol{u})} (r(\boldsymbol{u}) - s) = 0 \tag{49}$$

similarly for any $r_{kkk} \in R_+$ we have:

$$r_{kkk}(\boldsymbol{z}(\boldsymbol{u})) = \prod_{s \in \Omega_R(\boldsymbol{u})} (r_{kkk}(\boldsymbol{u}) + r_{kkk}(\hat{\boldsymbol{s}})) = \prod_{s \in \Omega_R(\boldsymbol{u})} (1 - s) \neq 0 \tag{50}$$

which is constant over different $k$. So $\boldsymbol{z}(\boldsymbol{u})$ satisfies Lemma 1, up to a scaling factor. $\qquad\square$

## C.2  CONSTRUCTION OF GLOBAL OPTIMIZERS

**Corollary 2** (Order-6 global optimizers). *The following "$3 \times 2$" Fourier solutions satisfy the sufficient condition (Lemma 1) and thus are global optimizers (assuming $d$ is odd):*

$$\boldsymbol{z}_{F6} = \frac{1}{\sqrt[3]{6}} \sum_{k=1}^{(d-1)/2} \boldsymbol{z}_{\mathrm{syn}}^{(k)} * \boldsymbol{z}_{\nu}^{(k)} * \boldsymbol{y}_k \tag{6}$$

*Here $\boldsymbol{z}_{\mathrm{syn}}^{(k)} := \boldsymbol{\rho}(\boldsymbol{u}_{\mathrm{syn}}^{(k)})$ and $\boldsymbol{z}_{\nu}^{(k)} := \boldsymbol{u}_{\nu}^{(k)} + \mathbf{1}_k$ (i.e., not maximal polynomial), where $\boldsymbol{u}_{\mathrm{syn}}$ and $\boldsymbol{u}_{\nu}$ are defined in Table 1. $\boldsymbol{y}$ is an order-1 unit. As a result, $\mathrm{ord}(\boldsymbol{z}_{F6}) = 3 \cdot 2 \cdot 1 \cdot (d-1)/2 = 3(d-1)$ and each frequency are affiliated with 6 hidden nodes (order-6).*

*Proof.* Just notice that $\boldsymbol{z}_{\mathrm{syn}} := \boldsymbol{\rho}(\boldsymbol{u}_{\mathrm{syn}}) = \boldsymbol{u}_{\mathrm{syn}}^2 + \boldsymbol{u}_{\mathrm{syn}} + \mathbf{1}_k$ (superscript $(k)$ are omitted for brevity) makes all MPs in $R_{\mathrm{n}}$, $R_{\mathrm{c}}$ and part of $R_*$ (Tbl. 1) equal to 0, except for "aac" and "bbc", which corresponds to monomial polynomials $r_{akkk} := \sum_j z_{akj} z_{akj} z_{ckj}$ and $r_{bkkk} := \sum_j z_{bkj} z_{bkj} z_{ckj}$. On the other hand, according to Tbl. 1, $\boldsymbol{z}_{\nu} := \boldsymbol{u}_{\nu} + \mathbf{1}_k$ has $r_{akkk}(\boldsymbol{z}_{\nu}) = r_{bkkk}(\boldsymbol{z}_{\nu}) = 0$. Therefore, using ring homomorphism, we know that for any $r \in R_{\mathrm{n}} \cup R_{\mathrm{c}} \cup R_*$, $r(\boldsymbol{z}_{\mathrm{syn}} * \boldsymbol{z}_{\nu}) = 0$ and thus $R_{\mathrm{n}} \cup R_{\mathrm{c}} \cup R_*$ is the 0-sets.

On the other hand for any $k'$, we have:

$$r_{k'k'k'}(\boldsymbol{z}_{F6}) = r_{k'k'k'}\left(\frac{1}{\sqrt[3]{6}} \sum_{k=1}^{(d-1)/2} \boldsymbol{z}_{\mathrm{syn}}^{(k)} * \boldsymbol{z}_{\nu}^{(k)} * \boldsymbol{y}_k\right) \tag{51}$$

$$= \frac{1}{6} \sum_{k=1}^{(d-1)/2} r_{k'k'k'}(\boldsymbol{z}_{\mathrm{syn}}^{(k)} * \boldsymbol{z}_{\nu}^{(k)} * \boldsymbol{y}_k) \tag{52}$$

$$= \frac{1}{6} \sum_{k=1}^{(d-1)/2} 6(\mathbb{I}(k = k') + \mathbb{I}(k = -k')) = 1 \tag{53}$$

The last equality is due to the fact that we only sum over half of the frequency. This means that $R_{\mathrm{g}}$ is a 1-set of $\boldsymbol{z}_{F6}$. Therefore, $\boldsymbol{z}_{F6}$ satisfies the sufficient condition (Eqn. 4) and the conclusion follows. $\qquad\square$

**Corollary 3** (Perfect Memorization). *We construct two $d$-order weights $\boldsymbol{z}_a$ and $\boldsymbol{z}_b$:*

$$\boldsymbol{z}_a = \sum_{j=0}^{d-1} \boldsymbol{u}_a^j, \qquad \boldsymbol{z}_b = \sum_{j=0}^{d-1} \boldsymbol{u}_b^j \tag{7}$$

Here $z_a \in R_c(k_1 \neq k) \cap R_n \cap R_*(p = b \text{ or } m \neq k)$, $z_b \in R_c(k_2 \neq k) \cap R_n \cap R_*(p = a \text{ or } m \neq k)$. Then $z_M = d^{-2/3} z_a * z_b$ satisfies the sufficient condition (Lemma 1) and is the perfect memorization solution with $\mathrm{ord}(z_M) = d^2$:

$$z_{akj_1j_2}^{(M)} = \omega^{kj_1}/\sqrt[3]{d^2}, \qquad z_{bkj_1j_2}^{(M)} = \omega^{kj_2}/\sqrt[3]{d^2}, \qquad z_{ckj_1j_2}^{(M)} = \omega^{-k(j_1+j_2)}/\sqrt[3]{d^2} \tag{8}$$

where each hidden node is indexed by $j = (j_1, j_2)$, $0 \leq j_1, j_2 < d$, $k \neq 0$.

*Proof.* Simply plugging in the solution and check whether the equations specified the equations. For $z_a$, for $k = 0$ everything is zero; for $k \neq 0$, we have:

$$r_{k_1k_2k}(z_a) = \sum_j a_{k_1j} b_{k_2j} c_{kj} = \sum_j \omega^{j(k_1-k)} = \mathbb{I}(k_1 = k \neq 0) \tag{54}$$

$$r_{amk'k}(z_a) = \sum_j a_{k'j} a_{m-k',j} c_{kj} = \sum_j \omega^{j(m-k)} = \mathbb{I}(m = k \neq 0) \tag{55}$$

$$r_{bmk'k}(z_a) = \sum_j b_{k'j} b_{m-k',j} c_{kj} = \sum_j \omega^{-jk} = \mathbb{I}(k = 0) = 0 \tag{56}$$

$$\tag{57}$$

Therefore, $z_a \in R_c(k_1 \neq k) \cap R_n \cap R_*(p = b \text{ or } m \neq k)$. Similar for $z_b$. For $z_M := d^{-2/3} z_a * z_b$, it satisfies all 0-sets constraints (i.e., for any $r$, either $z_a$ satisfies with $r(z_a) = 0$, or $z_b$ satisfies with $r(z_b) = 0$) and we have:

$$r_{kkk}(d^{-2/3} z_a * z_b) = d^{-2} r_{kkk}(z_a) r_{kkk}(z_b) = d^{-2} \cdot d \cdot d = 1 \tag{58}$$

So $z_M$ satisfies the sufficient conditions (Eqn. 4). $\qquad\square$

**Corollary 4** (Order-4 single frequency solution). *Define single frequency order-2 solution $z_\xi$:*

$$z_{ak\cdot} = [1, \xi], \quad z_{bk\cdot} = [1, -\mathrm{i}\bar{\xi}], \quad z_{ck\cdot} = [1, \mathrm{i}] \tag{9}$$

*where $|\xi| = 1$. Then the order-4 solution $z_{F4}^{(k)} := \rho(u_{\nu=\mathrm{i}}^{(k)}) * z_\xi^{(k)}$ has 0-sets $R_c$ and $R_*$ (but not $R_n$).*

*Proof.* First, $u_{\nu=\mathrm{i}} = u_{4c}$ in Tbl. 1 and thus $\rho(u_{\nu=\mathrm{i}})$ has 0-sets $R_c$ and $R_*$ except for "$ab\bar{c}$", which corresponds to MP $r_{k,k,-k} \in R_c$. On the other hand, we have

$$r_{k,k,-k}(z_\xi) = 1 + \xi \cdot (-\mathrm{i}\bar{\xi}) \cdot (-\mathrm{i}) = 0 \tag{59}$$

With the property of ring homomorphism, the conclusion follows. $\qquad\square$

**Corollary 5** (Mixed order-4/6 global optimizers). *With $z_{F4}^{(k)}$, there is a global optimizer to Eqn. 3 that does not meet the sufficient condition, i.e., $\sum_{k'} r_{p,k',-k',m} = 0$ but $r_{p,k',-k',m} \neq 0$:*

$$z_{F4/6} = \frac{1}{\sqrt[3]{6}} \hat{z}_{F6}^{(k_0)} + \frac{1}{\sqrt[3]{4}} \sum_{k=1, k \neq k_0}^{(d-1)/2} z_{F4}^{(k)} \tag{10}$$

*where $\hat{z}_{F6}^{(k_0)}$ is a perturbation of $z_{F6}^{(k_0)} := z_{\mathrm{syn}}^{(k_0)} * z_{\nu=1}^{(k_0)}$ by adding constant biases to its $(c, k)$ entries for $k \neq k_0$. The order is lower than $z_{F6}$: $\mathrm{ord}(z_{F4/6}) = 6 + 4 \cdot ((d-1)/2 - 1) = 2d < \mathrm{ord}(z_{F6})$.*

*Proof.* While $z_{F4}^{(k)}$ does not satisfy $R_n$, a weaker condition for a global optimizer to Theorem 1 is that $\sum_{k'} r_{p,k',-k',m} = 0$. We show that by adding constants to $(c, k)$ entries of $z_{F6}^{(k_0)}$ for $k \neq \pm k_0$, we can achieve that while not changing the value of other MPs.

To see this, we compute for each $m \neq \pm k_0$:

$$\sum_{k'} r_{p,k',-k',m}(\hat{z}_{F6}^{(k_0)}) = 2 \sum_{k'} \sum_j |[\hat{z}_{F6}^{(k_0)}]_{pk'j}|^2 [\hat{z}_{F6}^{(k_0)}]_{cmj} \tag{60}$$

$$= 2 \sum_j |[\hat{z}_{F6}^{(k_0)}]_{pk_0j}|^2 [\hat{z}_{F6}^{(k_0)}]_{cmj} = 2 \sum_j [\hat{z}_{F6}^{(k_0)}]_{cmj} \tag{61}$$

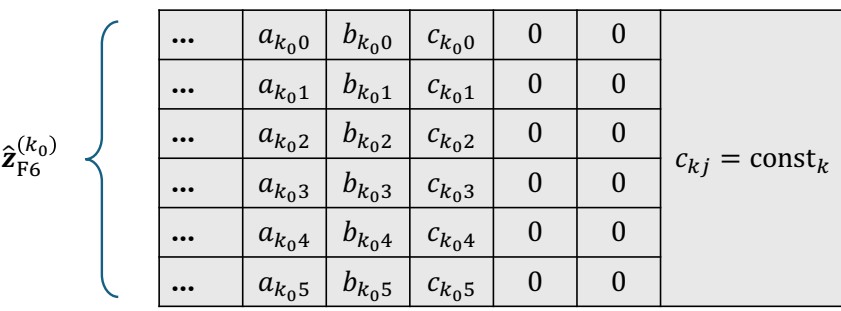

Figure 7: Visualization of $\hat{z}_{F6}^{(k_0)}$.

The second equality is because all $(a, k')$ and $(b, k')$ entries are 0 except for $k' = \pm k_0$, and the last equality is because all nonzero entries of $z_{F6}^{(k_0)}$ have magnitude 1.

On the other hand, we have:

$$\sum_{k'} r_{p,k',-k',m}\left(\sum_{k \neq k_0} z_{F4}^{(k)}\right) = \sum_{k'} r_{p,k',-k',m}(\rho(u_{4c}^{(m)}))r_{p,k',-k',m}(z_{\xi}^{(m)}) \tag{62}$$

$$= 2r_{p,m,-m,m}(\rho(u_{4c}^{(m)}))r_{p,m,-m,m}(z_{\xi}^{(m)}) \tag{63}$$

$$= 2(1+1)(1+i) = 4(1+i) \tag{64}$$

For $m = \pm k_0$, we have $r_{p,k',-k',m}(\hat{z}_{F6}^{(k_0)}) = 0$ and $r_{p,k',-k',m}(z_{F4}^{(k)}) = 0$ for $k \neq m$.

Therefore, we just let

$$[\hat{z}_{F6}^{(k_0)}]_{cmj} = -\frac{4(1+i)}{2 \cdot 6} = -\frac{1}{3}(1+i) \tag{65}$$

and $\sum_{k'} r_{p,k',-k',m}(z_{F4/6}) = 0$ for all $m$. See Fig. 7) for the construction.

To see why such a modification of $z_{F6}^{(k_0)}$ won't change other MPs, simply notice that candidate MPs that may not be zero anymore are $r_{\pm k_0 \pm k_0 m}$, $r_{pk_0 k_0 m}$ and $r_{p,-k_0,-k_0,m}$ for $m \neq \pm k_0$. For $m = \pm k_0$, $z_{F6}^{(k_0)}$ are well behaved.

Note that $r_{\pm k_0 \pm k_0 k}(\hat{z}_{F6}^{(k_0)})$ is the same as applying $r_{\pm k_0 \pm k_0 k_0}$ to a solution $\hat{z}$ which replaces $(c, k_0)$ entries of $\hat{z}_{F6}^{(k_0)}$ by $(c, m)$ entries. Let $\hat{u}_{syn} = [\omega_3, \omega_3, 1]$ and $\hat{u}_{one} = [1, -1, 1]$. Then $\hat{z} = \rho(\hat{u}_{syn}) * \rho(\hat{u}_{one})$ and thus for $m \neq \pm k_0$, we have:

$$r_{\pm k_0 \pm k_0 m}(z_{F4/6}) = r_{\pm k_0 \pm k_0 m}(\hat{z}_{F6}^{(k_0)}) \propto r_{\pm k_0 \pm k_0 k_0}(\hat{z}) \tag{66}$$

$$= r_{\pm k_0 \pm k_0 k_0}(\rho(\hat{u}_{syn}))r_{\pm k_0 \pm k_0 k_0}(\rho(\hat{u}_{one})) = 0 \tag{67}$$

since $r_{\pm k_0 \pm k_0 k_0}(\rho(\hat{u}_{one})) = 0$. Similarly for $m \neq \pm k_0$,

$$r_{pk_0 k_0 m}(z_{F4/6}) = r_{pk_0 k_0 m}(\hat{z}_{F6}^{(k_0)}) \propto r_{pk_0 k_0 k_0}(\hat{z}) \tag{68}$$

$$= r_{pk_0 k_0 k_0}(\rho(\hat{u}_{syn}))r_{pk_0 k_0 k_0}(\rho(\hat{u}_{one})) = 0 \tag{69}$$

since $r_{pk_0 k_0 k_0}(\rho(\hat{u}_{syn})) = 0$. Similarly for $r_{p,-k_0,-k_0,m}$. $\qquad\square$

### C.3 CANONICAL FORMS

**Definition 8.** *A solution $z$ is called* canonical at $k_0$, *or $z \in \mathcal{C}_{k_0}$, if $z_{pk0} = 1$ for all $p$ and $k = \pm k_0$.*

**Lemma 4** (Canonical Decomposition). *Any solution $z$ with $r_{k_0 k_0 k_0}(z) \neq 0$ can be decomposed into $z = z' * y$, where $z'$ is canonical at $k_0$ and $\text{ord}(y) = 1$. Both $r_{k_0 k_0 k_0}(z') \neq 0$ and $r_{k_0 k_0 k_0}(y) \neq 0$.*

*Proof.* Since $r_{k_0k_0k_0}(\boldsymbol{z}) = \sum_j a_{k_0j}b_{k_0j}c_{k_0j} \neq 0$, there must exist some $j$ so that $z_{ak_0j}z_{bk_0j}z_{ck_0j} \neq 0$, which means that $z_{ak_0j} \neq 0$, $z_{bk_0j} \neq 0$ and $z_{ck_0j} \neq 0$. Since the node index $j$ can be permuted, we can let node $j$ be the first node $0$ and let $y_{pk0} = z_{pkj}$ and $z'_{pkj'} = z_{pkj'}z_{pkj}^{-1}$ for $p \in \{a, b, c\}$ and $k = \pm k_0$, then $\boldsymbol{z}'$ is canonical at $k_0$ and $\mathrm{ord}(\boldsymbol{y}) = 1$. Finally, by ring homomorphism, since

$$r_{k_0k_0k_0}(\boldsymbol{z}) = r_{k_0k_0k_0}(\boldsymbol{z}')r_{k_0k_0k_0}(\boldsymbol{y}) \neq 0 \tag{70}$$

we know that both $r_{k_0k_0k_0}(\boldsymbol{z}') \neq 0$ and $r_{k_0k_0k_0}(\boldsymbol{y}) \neq 0$. $\qquad\square$

**Lemma 5** (Necessary Condition for $R_c$). *All order-1 and order-2 solutions satisfying $R_c := \{r_{k_1k_2k} = 0, k_1, k_2, k \text{ not all equal}\}$ must have $r_{kkk} = 0$ for all $k$ (i.e. the first equation in Eqn. 4 cannot be satisfied).*

*Proof.* For any order-1 solution, for any $k$, in order to make $r_{k,-k,k} = z_{ak0}z_{b,-k,0}z_{ck0} = z_{ak0}\bar{z}_{bk0}z_{ck0} = 0$, either $z_{ak0}$, $z_{bk0}$ or $z_{ck0}$ has to be zero, which means that $r_{kkk} = 0$.

For order-2, first of all if any $z_{pk0} = 0$ for any $p \in \{a, b, c\}$, then a constraint like $r_{k,k,-k} = z_{ak0}z_{bk0}\bar{z}_{ck0} + z_{ak1}z_{bk1}\bar{z}_{ck1} = 0$ yields $z_{ak1}z_{bk1}z_{ck1} = 0$ and thus $r_{kkk} = 0$. If not, then for any two complex numbers $z_{pk0}$ and $z_{pk1}$, there always exist four real numbers $\theta_p \in (-\pi, \pi]$, $\theta'_p \in (-\pi, \pi]$, $m_{p0} > 0$ and $m_{p1} > 0$ so that

$$z_{pk0} = m_{p0}e^{\mathrm{i}\theta'_p}e^{\mathrm{i}\theta_p}, \qquad z_{pk1} = m_{p1}e^{\mathrm{i}\theta'_p}e^{-\mathrm{i}\theta_p} \tag{71}$$

Then a constraint like $r_{k,k,-k} = z_{ak0}z_{bk0}\bar{z}_{ck0} + z_{ak1}z_{bk1}\bar{z}_{ck1} = 0$ can be written as $z_{ak0}z_{bk0}\bar{z}_{ck0} = -z_{ak1}z_{bk1}\bar{z}_{ck1}$, or equivalently:

$$m_{a0}m_{b0}m_{c0}e^{\mathrm{i}(\theta'_a+\theta'_b+\theta'_c)}e^{\mathrm{i}(\theta_a+\theta_b-\theta_c)} = -m_{a1}m_{b1}m_{c1}e^{\mathrm{i}(\theta'_a+\theta'_b+\theta'_c)}e^{-\mathrm{i}(\theta_a+\theta_b-\theta_c)} \tag{72}$$

$$m_{a0}m_{b0}m_{c0}e^{\mathrm{i}\theta_a}e^{\mathrm{i}\theta_b}e^{-\mathrm{i}\theta_c} = -m_{a1}m_{b1}m_{c1}e^{-\mathrm{i}\theta_a}e^{-\mathrm{i}\theta_b}e^{\mathrm{i}\theta_c} \tag{73}$$

Comparing their magnitude and phase, we have $m_{a0}m_{b0}m_{c0} = m_{a1}m_{b1}m_{c1}$ and

$$\theta_a + \theta_b - \theta_c = \pm\pi/2 \mod 2\pi \tag{74}$$

Similarly, we have:

$$\theta_a + \theta_c - \theta_b = \pm\pi/2 \mod 2\pi, \qquad \theta_b + \theta_c - \theta_a = \pm\pi/2 \mod 2\pi \tag{75}$$

Solving the three equations and we have 6 possible solutions:

$$(\theta_a, \theta_b, \theta_c) = (0, 0, \pm\pi/2) \mod 2\pi \tag{76}$$

$$(\theta_a, \theta_b, \theta_c) = (0, \pm\pi/2, 0) \mod 2\pi \tag{77}$$

$$(\theta_a, \theta_b, \theta_c) = (\pm\pi/2, 0, 0) \mod 2\pi \tag{78}$$

For all such solutions, let $m := m_{a0}m_{b0}m_{c0} = m_{a1}m_{b1}m_{c1}$, then we have:

$$r_{kkk} = z_{ak0}z_{bk0}z_{ck0} + z_{ak1}z_{bk1}z_{ck1} \tag{79}$$

$$= me^{\mathrm{i}(\theta'_a+\theta'_b+\theta'_c)}(e^{\mathrm{i}(\theta_a+\theta_b+\theta_c)} + e^{-\mathrm{i}(\theta_a+\theta_b+\theta_c)}) \tag{80}$$

$$= me^{\mathrm{i}(\theta'_a+\theta'_b+\theta'_c)}(e^{\pm\mathrm{i}\pi/2} + e^{\mp\mathrm{i}\pi/2}) \tag{81}$$

$$= 0 \tag{82}$$

$\qquad\square$

**Lemma 6** (Property of order-3 solutions satisfying $R_c$ and $R_g$). *With small $L_2$ regularization, all per-frequency order-3 canonical solutions $\boldsymbol{z}$ at frequency $k_0$ that satisfy $R_c$ and $R_g$ are in the following form:*

$$z_{pk_0\cdot} = [1, \alpha_p\omega_3, \beta_p\bar{\omega}_3], \qquad \text{for } p \in \{a, b, c\} \tag{83}$$

*where $\alpha_p = \pm 1$ and $\beta_p = \pm 1$ with the constraint that $\alpha_a\alpha_b\alpha_c = \beta_a\beta_b\beta_c = 1$. For $k \neq k_0, z_{pk\cdot} = 0$.*

*Proof.* We first prove that $\boldsymbol{z}$ satisfies $R_c$ and $R_g$. To see this, we have

$$r_{k_1 k_2 k} = \sum_j \mathbb{I}(k_1 = k_2 = k = k_0)\omega_3^{3j} + \sum_j \mathbb{I}(-k_1 = k_2 = k = k_0)\omega_3^j \tag{84}$$

$$+ \ldots + \sum_j \mathbb{I}(-k_1 = -k_2 = -k = k_0)\bar{\omega}_3^{3j} \tag{85}$$

$$= 3\mathbb{I}(k_1 = k_2 = k = k_0) + 3\mathbb{I}(k_1 = k_2 = k = -k_0) \tag{86}$$

Note that all cross terms are gone since $\sum_j \omega_3^j = 0$. It is clear that $r_{k_1 k_2 k} \neq 0$ unless $k_1 = k_2 = k$ so $\boldsymbol{z}$ satisfies $R_c$ and $R_g$.

Now we consider any per-frequency order-3 canonical solution (Def. 8) at frequency $k$. Let $a_j := z_{akj}$, $b_j := z_{bkj}$ and $c_j := z_{ckj}$. Let $\boldsymbol{a} = [a_j] \in \mathbb{C}^3$, $\boldsymbol{b} = [b_j] \in \mathbb{C}^3$ and $\boldsymbol{c} = [c_j] \in \mathbb{C}^3$. Since the solution is canonical, we have $a_0 = b_0 = c_0 = 1$.

Then the conditions yield that

$$(\boldsymbol{a} \circ \bar{\boldsymbol{b}})^\top \boldsymbol{c} = 0, \quad (\boldsymbol{a} \circ \bar{\boldsymbol{b}})^\top \bar{\boldsymbol{c}} = 0, \quad (\bar{\boldsymbol{a}} \circ \boldsymbol{b})^\top \boldsymbol{c} = 0, \quad (\bar{\boldsymbol{a}} \circ \boldsymbol{b})^\top \bar{\boldsymbol{c}} = 0 \tag{87}$$

which means that in $\mathbb{R}^3$ space, the following condition holds:

$$\text{span}(\Re(\boldsymbol{a} \circ \bar{\boldsymbol{b}}), \Im(\boldsymbol{a} \circ \bar{\boldsymbol{b}})) \perp \text{span}(\Re(\boldsymbol{c}), \Im(\boldsymbol{c})) \tag{88}$$

where $\Re(\cdot)$ and $\Im(\cdot)$ are real and imaginary parts of a complex vector. Since Eqn. 88 holds in $\mathbb{R}^3$, it must be the following cases: either $\Re(\boldsymbol{a} \circ \bar{\boldsymbol{b}})$ is co-linear with $\Im(\boldsymbol{a} \circ \bar{\boldsymbol{b}})$, or $\Re(\boldsymbol{c})$ is co-linear with $\Im(\boldsymbol{c})$.

If the latter is true (i.e., there exists $\beta$ so that $\beta\Re(\boldsymbol{c}) = \Im(\boldsymbol{c})$), then since $c_0 = 1$ is real, $\beta = 0$ and $\Im(\boldsymbol{c}) = 0$. So $\boldsymbol{c}$ is real. In this case,

$$r_{kkk} = (\boldsymbol{a} \circ \boldsymbol{b})^\top \boldsymbol{c} = (\boldsymbol{a} \circ \boldsymbol{b})^\top \bar{\boldsymbol{c}} = 0 \tag{89}$$

If the former is true, then similarly we conclude that $\Im(\boldsymbol{a} \circ \bar{\boldsymbol{b}}) = 0$ and $\boldsymbol{a} \circ \bar{\boldsymbol{b}}$ is real. Applying the same reasoning symmetrically, in order to find cases such that $r_{kkk} \neq 0$, a necessary condition is that

$$\boldsymbol{a} \circ \bar{\boldsymbol{b}}, \boldsymbol{b} \circ \bar{\boldsymbol{c}}, \boldsymbol{c} \circ \bar{\boldsymbol{a}} \in \mathbb{R}^3 \tag{90}$$

Let $z_{pkj} = |z_{pkj}|e^{\mathrm{i}\theta_{pj}}$. Let's first consider the case that $\boldsymbol{a} \circ \bar{\boldsymbol{b}}, \boldsymbol{b} \circ \bar{\boldsymbol{c}}, \boldsymbol{c} \circ \bar{\boldsymbol{a}} \in \mathbb{R}^3_{\geq 0}$. Then we have $\theta_{a0} = \theta_{b0} = \theta_{c0} = \theta_0 = 0$, $\theta_{a1} = \theta_{b1} = \theta_{c1} = \theta_1$, $\theta_{a2} = \theta_{b2} = \theta_{c2} = \theta_2$. Letting $m_j := |a_j||b_j||c_j|$, then the corresponding $r_{kkk}$ can be written as:

$$r_{kkk} = \sum_{j=0}^2 m_j e^{3\mathrm{i}\theta_j} \tag{91}$$

with the constraints that $\sum_{j=0}^2 m_j e^{\mathrm{i}\theta_j} = 0$ imposed by $R_c$.

**Minimal Norm solutions**. One interesting question is that what is the minimal norm representation that achieves the highest objective? For this we can solve the following optimization problem:

$$\max_{\{m_j, \theta_j\}} \sum_j m_j(e^{3\mathrm{i}\theta_j} + e^{-3\mathrm{i}\theta_j}) - \epsilon \sum_j m_j^2 \quad \text{s.t.} \quad \sum_j m_j e^{\mathrm{i}\theta_j} = 0 \tag{92}$$

which achieves the maximal when $m_j = 1/\epsilon$, $\theta_1 = 2\pi\mathrm{i}/3$ and $\theta_2 = 4\pi\mathrm{i}/3$ (or vise versa). Note that the optimal $\theta_j$ is fixed no matter how small the regularization coefficient $\epsilon$ is.

To see that, let $u_j := e^{\mathrm{i}\theta_j}$. Then we have:

$$\sum_j m_j(u_j + \bar{u}_j)^3 = \sum_j m_j[u_j^3 + 3u_j\bar{u}_j(u_j + \bar{u}_j) + \bar{u}_j^3] = \sum_j m_j(u_j^3 + \bar{u}_j^3) \tag{93}$$

Therefore, letting $x_j := 2\Re u_j$, we just need to consider the real part of the objective, and solve the following optimization in $\mathbb{R}$:

$$\max_{\{m_j, -2 \leq x_j \leq 2, x_0 = 2\}} \sum_j m_j x_j^3 - \epsilon \sum_j m_j^2 \quad \text{s.t.} \quad \sum_j m_j x_j = 0 \tag{94}$$

whose solutions give a sufficient condition. Using Lagrangian multiplier, we have:

$$\frac{\partial L}{\partial x_j} = m_j(3x_j^2 - \lambda) = 0, \qquad \frac{\partial L}{\partial m_j} = x_j^3 - 2\epsilon m_j - \lambda x_j = 0 \qquad (95)$$

which leads to $\lambda = 3$, $m_j = 1/\epsilon$ and $x_1 = x_2 = -1$. This corresponds to the solution

$$z_{pk\cdot} = [1, \omega_3, \bar{\omega}_3], \qquad \text{where } p \in \{a, b, c\} \qquad (96)$$

Note that the original necessary condition is $\boldsymbol{a} \circ \bar{\boldsymbol{b}}, \boldsymbol{b} \circ \bar{\boldsymbol{c}}, \boldsymbol{c} \circ \bar{\boldsymbol{a}} \in \mathbb{R}^3$. Considering the possible negativity, the solutions can be written as

$$z_{pk\cdot} = [1, \alpha_p \omega_3, \beta_p \bar{\omega}_3], \qquad \text{for } p \in \{a, b, c\} \qquad (97)$$

where $\alpha_p = \pm 1$ and $\beta_p = \pm 1$ with the constraint that $\alpha_a \alpha_b \alpha_c = \beta_a \beta_b \beta_c = 1$. $\qquad \square$

**Remarks.** Note that this conclusion does not contradict with the constructed solution $\boldsymbol{z}_{\text{syn},\alpha\beta}$ in Eqn. 11 in which $\alpha$ and $\beta$ are allowed to be any complex number with magnitude 1. This is because $\boldsymbol{z}_{\text{syn},\alpha\beta}$ does not satisfy all the constraints in $R_c$ (but $\boldsymbol{z}_{\text{syn},\alpha\beta} * \boldsymbol{z}_{\nu=\text{i}}$ will) unless $\alpha$ and $\beta$ are real and thus $\pm 1$.

## D  GRADIENT DYNAMICS (PROOF)

**Theorem 5** (Dynamics of MPs). *The dynamics of MPs satisfies $\dot{\boldsymbol{r}} = -JJ^*\overline{\nabla_{\boldsymbol{r}}\ell}$, which has positive inner product with the negative gradient direction $-\overline{\nabla_{\boldsymbol{r}}\ell}$.*

*Proof.* By gradient descent of $\mathcal{W}$, we have $\dot{\mathcal{W}} = -\overline{\nabla_{\mathcal{W}}\ell}$. By chain rule, we have:

$$\dot{\mathcal{W}} = -\overline{\nabla_{\mathcal{W}}\ell} = -\overline{J^\top \nabla_{\boldsymbol{r}}\ell} = -J^*\overline{\nabla_{\boldsymbol{r}}\ell} \qquad (98)$$

Then the dynamics of $\boldsymbol{r} = \boldsymbol{r}(\boldsymbol{z}(\mathcal{W}))$, as driven by the dynamics of $\mathcal{W}$, is given by

$$\dot{\boldsymbol{r}} = J\dot{\mathcal{W}} = -JJ^*\overline{\nabla_{\boldsymbol{r}}\ell} \qquad (99)$$

To show positive inner product, we have:

$$-\overline{\nabla_{\boldsymbol{r}}\ell}^* \dot{\boldsymbol{r}} = \overline{\nabla_{\boldsymbol{r}}\ell}^* JJ^*\overline{\nabla_{\boldsymbol{r}}\ell} = \|J^*\overline{\nabla_{\boldsymbol{r}}\ell}\|_2^2 \geq 0 \qquad (100)$$

$\qquad \square$

**Theorem 6** (The Occam's Razer: Preference of low-order solutions). *If $\boldsymbol{z} = \boldsymbol{y} * \boldsymbol{z}'$ and both $\boldsymbol{z}$ (of order $q$) and $\boldsymbol{z}'$ are global optimal solutions, then there exists a path of zero loss connecting $\boldsymbol{z}$ and $\boldsymbol{z}'$ in the space of $\mathcal{Z}_q$. As a result, lower-order solutions are preferred if trained with $L_2$ regularization.*

*Proof.* Let $\text{ord}(\boldsymbol{z}) = q$ and $\text{ord}(\boldsymbol{z}') = q'$. Then $q'|q$. Since both $\boldsymbol{z}$ and $\boldsymbol{z}'$ are global optimal. Since $r_{kkk}$ is ring homomorphism, we know that $r_{kkk}(\boldsymbol{z}) = r_{kkk}(\boldsymbol{z}')r_{kkk}(\boldsymbol{y}) = 1/2d = r_{kkk}(\boldsymbol{z}')$ and thus $r_{kkk}(\boldsymbol{y}) = 1$ for all $k \neq 0$.

Let the augmented identity $\boldsymbol{e} \in \mathcal{Z}_q$ be $e_{pmj} = \mathbb{I}(j = 0)$. Then $r_{kkk}(\boldsymbol{e}) = 1$ for all $k \neq 0$.

We want to construct a path in $\mathcal{Z}_q$, the space of order-$q$ solutions as follows:

$$\tilde{\boldsymbol{z}}(t) = \tilde{\boldsymbol{y}}(t) * \boldsymbol{z}', \qquad 0 \leq t \leq 1 \qquad (101)$$

in which $\tilde{\boldsymbol{y}}(0) = \boldsymbol{e}$, $\tilde{\boldsymbol{y}}(1) = \boldsymbol{y}$, and $r_{kkk}(\tilde{\boldsymbol{y}}(t)) = 1$ for any $t$. To see why this is possible, pick a continuous family of trajectories $\hat{\boldsymbol{y}}(t; \lambda)$ with $\lambda \in [0, 1]$ so that they satisfies

$$\hat{\boldsymbol{y}}(0; \lambda) = \boldsymbol{e}, \qquad \hat{\boldsymbol{y}}(1; \lambda) = \boldsymbol{y}, \qquad r_{kkk}(\hat{\boldsymbol{y}}(t; 0)) \leq 1, \qquad r_{kkk}(\hat{\boldsymbol{y}}(t; 1)) \leq 1 \qquad (102)$$

which can always be achieved by scaling some trajectory with a factor that depends on $\lambda$. Then by intermediate theorem, there exists $\lambda(t)$ so that $r_{kkk}(\hat{\boldsymbol{y}}(t; \lambda(t))) = 1$ for some $k$. Note that for different frequency $k$ and $k'$, $r_{kkk}$ and $r_{k'k'k'}$ involves disjoint components of $\boldsymbol{z}$ so we could find such a path for all $k \neq 0$.

Therefore, for any monomial potential $r$ included in MSE loss (Eqn. 3), we have

$$r(\tilde{z}(t)) = r(\tilde{y}(t))r(z') = \begin{cases} \text{finite} \cdot 0 = 0 & r \neq r_{kkk} \\ 1 \cdot 1/2d = 1/2d & r = r_{kkk} \end{cases} \tag{103}$$

and thus the entire trajectory $\tilde{z}(t) = \tilde{y}(t) * z' \in \mathcal{Z}_q$ connecting $z$ and $e * z'$, which is $z'$ in the space of $\mathcal{Z}_q$, is also globally optimal.

To see why weight decay regularization leads to lower-order solution, we could simply compare the $\ell_2$ norm of $z = y * z'$ and $e * z'$. At each frequency $k$, this reduces to the following optimization problem:

$$\min \sum_j |a_j|^2 + |b_j|^2 + |c_j|^2, \qquad \text{s.t.} \sum_j a_j b_j c_j = 1 \tag{104}$$

where $a_j := y_{akj}$, $b_j := y_{bkj}$ and $c_j := y_{ckj}$. Since we know that arithmetic mean is no less than geometric mean:

$$\frac{|a_j|^2 + |b_j|^2 + |c_j|^2}{3} \geq \sqrt[3]{|a_j b_j c_j|^2} \tag{105}$$

We have:

$$\sum_j |a_j|^2 + |b_j|^2 + |c_j|^2 \geq 3 \sum_j |a_j b_j c_j|^{2/3} \geq 3 \tag{106}$$

The last inequality holds because (1) if any $|a_j b_j c_j| \geq 1$, then it holds, (2) if all $|a_j b_j c_j| < 1$, then since $a^x$ is a decreasing function for $a < 1$, $\sum_j |a_j b_j c_j|^{2/3} \geq \sum_j |a_j b_j c_j| \geq |\sum_j a_j b_j c_j| = 1$.

The minimizer is reached when $|a_j| = |b_j| = |c_j|$. Note that if $a_j b_j c_j$ has any complex phase or negative, then in order to satisfy $\sum_j a_j b_j c_j = 1$, objective function needs to be larger. So without loss of generality, we could study $a_j = b_j = c_j = x_j \geq 0$ and the optimization problem becomes

$$\min \sum_j x_j^2, \qquad \text{s.t.} \sum_j x_j^3 = 1, \quad x_j \geq 0 \tag{107}$$

which has a minimizer at the corners $(1, 0, \ldots)$. This corresponds to $a_j = b_j = c_j = \mathbb{I}(j = 0)$, which is the augmented identity $e \in \mathcal{Z}_q$. $\qquad \square$

**Theorem 7** (Infinite Width Limits at Initialization). *Considering the modified loss of Eqn. 3 with only the first two terms:* $\tilde{\ell}_k := -2r_{kkk} + \sum_{k_1 k_2} |r_{k_1 k_2 k}|^2$, *if the weights are i.i.d Gaussian and network width* $q \to +\infty$, *then* $JJ^*$ *converge to diagonal and the dynamics of MPs is decoupled.*

*Proof.* Let $\tilde{\ell} := \sum_k \nabla \tilde{\ell}_k$. Let's compute the dynamics of MPs following Theorem 5: $\dot{r} = -JJ^* \overline{\nabla_r \tilde{\ell}}$.

First it is clear that

$$\frac{\partial \tilde{\ell}}{\partial r_{k_1 k_2 k}} = \sum_k \frac{\partial \tilde{\ell}_k}{\partial r_{k_1 k_2 k}} = -2\mathbb{I}(k_1 = k_2 = k) + 2\overline{r_{k_1 k_2 k}} \tag{108}$$

So the $(k_1, k_2, k)$ component of $\overline{\nabla_r \tilde{\ell}}$ only contains $r_{k_1 k_2 k}$.

Then we compute $H := JJ^*$ and show that it is asymptotically diagonal. To see this, each component of $H$, i.e., $h_{k_1 k_2 k_3, k_1' k_2' k_3'}$ can be computed as the following:

$$h_{k_1 k_2 k_3, k_1' k_2' k_3'} = \sum_{pmj} \frac{\partial r_{k_1 k_2 k_3}}{\partial z_{pmj}} \overline{\frac{\partial r_{k_1' k_2' k_3'}}{\partial z_{pmj}}} \tag{109}$$

$$= \mathbb{I}(k_1 = k_1') \sum_j b_{k_2 j} \bar{b}_{k_2' j} c_{k_3 j} \bar{c}_{k_3' j} \tag{110}$$

$$+ \mathbb{I}(k_2 = k_2') \sum_j a_{k_1 j} \bar{a}_{k_1' j} c_{k_3 j} \bar{c}_{k_3' j} \tag{111}$$

$$+ \mathbb{I}(k_3 = k_3') \sum_j a_{k_1 j} \bar{a}_{k_1' j} b_{k_2 j} \bar{b}_{k_2' j} \tag{112}$$

where $a_{kj} := z_{akj}$, $b_{kj} := z_{bkj}$ and $c_{kj} := z_{ckj}$. Then for component $(k_1 k_2 k_3, k_1', k_2', k_3')$, if any $k_p \neq k_p'$ for some $p \in \{a, b, c\}$, then the corresponding $z_{pk_p j} \bar{z}_{pk_p' j}$ has random phase for hidden node $j$, and $h_{k_1 k_2 k_3, k_1' k_2' k_3'} \to 0$ when $q \to +\infty$.

Combining the two, we know that the dynamics of MPs is decoupled, that is, each $r_{k_1 k_2 k}$ evolves independently over time. $\qquad \square$

**Ripple effects**. While Theorem 7 only holds at initialization, the resulting decoupled MP dynamics, e.g., $dr_{kkk}/dt = 1 - r_{kkk}$ that leads to $r_{kkk}(t) = 1 - e^{-t}$, already captures the rough shape of the curve (Fig. 3 top right). To capture its fine structures (e.g., ripples before stabilization), we can also model the dynamics of the diagonal element in $JJ^*$. Consider a symmetric 1D case on a fixed frequency $k$, where all diagonal $r_{kkk} = r_0 - r$ (where $r_0 = 1/2d$) and all off-diagonal $r_{k_1 k_2 k} = r$, then

$$\dot{r} = -\dot{r}_{kkk} = \kappa(r_{kkk} - r_0) = -\kappa r, \quad \dot{\kappa} = \alpha(r_0 - r_{kkk}) - (1 - \alpha)r_{k_1 k_2 k} - c_0 = (2\alpha - 1)r - c_0 \quad (113)$$

where $\kappa > 0$ is the diagonal element of $JJ^*$ and $\alpha$ is a coefficient that characterizes the relative strength of two negative gradient $-\overline{\nabla_{r_{kkk}}}\ell = r_0 - r_{kkk}$ and $-\overline{\nabla_{r_{k_1 k_2 k}}}\ell = -r_{k_1 k_2 k}$, and $c_0$ is the gradient terms caused by asymmetry and/or other frequencies. This yields a second-order ODE that has complex roots in the characteristic function when $c_0 > 0$.

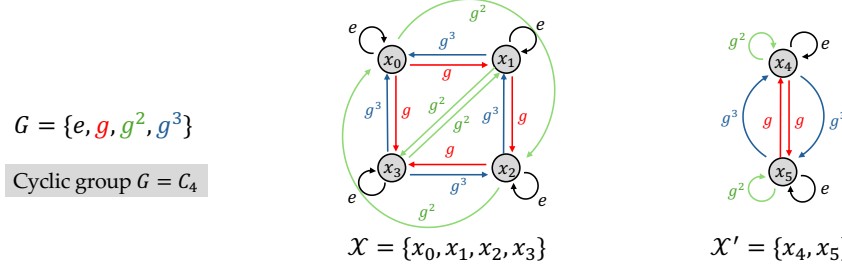

$$G = \{e, g, g^2, g^3\}$$

Cyclic group $G = C_4$

$$\mathcal{X} = \{x_0, x_1, x_2, x_3\} \qquad \mathcal{X}' = \{x_4, x_5\}$$

Figure 8: An example case of group action on state set $\mathcal{X}$, $\mathcal{X}$ can be partitioned into several disjointed components, each is a transitive graph w.r.t the group actions in $G$.

# E    EXTENDING CoGO TO GROUP ACTION PREDICTION

While in this work we mainly focus on Abelian group, CoGO can be extended to more general *group action prediction*: given a group element $g \in G$ and the current state $x \in \mathcal{X}$, the goal is to predict $gx \in X$, i.e., the next state after action $g$. Such tasks include modular addition/multiplication in which the group acts on itself (i.e., $\mathcal{X} = G$), and also includes the transition function in reinforcement learning (Sutton, 2018) and world modeling (Garrido et al., 2024), in which an action changes the current state to a new one.

**Setup**. Consider a state space $\mathcal{X}$ and *group action* $G \times \mathcal{X} \mapsto \mathcal{X}$ where $g \in G$ is a group element acting on a state $x \in \mathcal{X}$ to get an update state $gx \in \mathcal{X}$. It satisfies two axioms (1) the group identity maps everything to itself: $ex = x$, and (2) the group action is compatible with group multiplication: $g(hx) = (gh)x$ for any $g, h \in G$ and $x \in \mathcal{X}$.

Equipped with the group action, the state space now can be decoupled into a disjoint of *transitive components*.

**Definition 9** (Transitive group action). *A group action is transitive, if for any $x_1, x_2 \in \mathcal{X}$, there exists $g \in G$ so that $gx_1 = x_2$.*

Since the group action is compatible with multiplication, $\mathcal{X}$ under $G$ will be partitioned into disjoint components $\mathcal{X} = \bigcup_l \mathcal{X}_l$ and we can analyze each component separately (Fig. 8).

**Transitive Group Action**. For each transitive component $\mathcal{X}$ (dropping $l$ for brevity), under certain conditions, we could define a *state multiplication* operation (a formal definition in Def. 10 in Appendix) so that for any group action $gx \in \mathcal{X}$, there is an associated state $x' \in \mathcal{X}$ so that $x' \cdot x = gx$. Furthermore, under the multiplication, $\mathcal{X}$ itself becomes a group:

**Theorem 8** ($\mathcal{X} \cong G/G_{x_0}$). *If the group stabilizer $G_{x_0} := \{g | gx_0 = x_0\}$ is a normal subgroup of $G$, then $\mathcal{X}$ is isomorphic to the quotient group $G/G_{x_0}$ and thus forms a group.*

Moreover, we can prove that for any group element $g \in G$, there exists $x = \iota_0(g) \in \mathcal{X}$ so that for any state $x'$, the group action $gx'$ is the same as the state multiplication $x' \cdot x$. Therefore, for group action prediction tasks, we have (note the difference compared to Eqn. 12):

$$\mathbf{w}_j = U_G \left( P_0 \mathbf{w}_{j,G}^{||} + \mathbf{w}_{j,G}^{\perp} \right) + U_{\mathcal{X}} \mathbf{w}_{j,\mathcal{X}} \tag{114}$$

where $\mathbf{w}_{j,G}^{||} \in \mathbb{R}^{|\mathcal{X}|}$ is the "in-graph" component of $G$, $\mathbf{w}_{j,G}^{\perp} \in \mathbb{R}^{|G|}$ is the "out-of-graph" component of $G$, and $P_0 \in \mathbb{R}^{|G| \times |\mathcal{X}|}$ "lifts" from $\mathcal{X}$ to $G$ using $\iota_0$, i.e., $(P_0)_{gx} = 1$ for $g \in \iota_0^{-1}(x)$, and $\mathbf{w}_{j,G}^{\perp} \perp P_0 \mathbf{w}_{j,G}^{||}$. Since any $g$ just behaves like $\iota_0(g)$ when acting on $\mathcal{X}$, our framework can be applied to characterize the learning of $\mathbf{w}_{j,G}^{||}$. Intuitively, we only learn representation of $G$'s element "module" its kernel $G_{x_0}$, since element in the kernel is indistinguishable from each other.

On the other hand, the behavior of $\mathbf{w}_{j,G}^{\perp}$ will be influenced by $g$ acting on other graphs, and the final learned representation of a group element $g$ is the direct sum of them.

# F  DETAILED EXPLANATION OF SEC. E

**Matrix Representation**. Each group element $g$ can be represented by a matrix $R_g$, i.e., its *matrix representation*, so that it respects the group multiplication (i.e., *homomorphism*): $R_{gh} = R_g R_h$ for any group elements $g, h \in G$.

The dimension of such a representation may differ widely. Some representation can be 1-dimensional (e.g., for Abelian group), while others can be infinitely dimensional. The *permutation representation* $R_g \in \mathbb{R}^{d \times d}$ maps a one-hot representation $e_x \in \mathbb{R}^d$ of an object $\mathcal{X}$ into its image $e_{gx} \in \mathbb{R}^d$, also a one-hot representation. Intuitively, $(R_g)_{jk} = 1$ means that it maps the $k$-th element into the $j$-th element.

**Lemma 7** (Structure of $R_g$). *For any $g \in G$, $R_g$ is a permutation matrix.*

**Lemma 8** (Summation of $R_g$). *If the group action is transitive, then $\sum_{g \in G} R_g = \frac{|G|}{d} \mathbf{1}\mathbf{1}^\top$.*

## F.1  TRANSITIVE CASE

To construct the multiplication operation on $\mathcal{X}$, we first pick reference point $x_0 \in \mathcal{X}$, and establish a mapping $\iota_0 : G \mapsto \mathcal{X}$: $\iota_0(g) = gx_0$. Note that $\iota_0$ is not necessarily a bijection; in fact we have:

**Lemma 9** (Co-set Mapping $\iota_0$). *There is a bijection between $\{\iota_0^{-1}(x)\}_{x \in \mathcal{X}}$ and co-sets $[G : G_{x_0}]$ of group stabilizer $G_{x_0} := \{g \in G | gx_0 = x_0\}$, which is a subgroup of $G$ fixing $x_0$.*

**Lemma 10** (Uniqueness of Multiplication Mapping). *If $G_{x_0}$ is a normal subgroup, then for all $g_1 \in \iota_0^{-1}(x_1)$ and $g_2 \in \iota_0^{-1}(x_2)$, all $g_1 g_2 G_{x_0}$ correspond to the same coset.*

**Definition 10** (The multiplication operator on $\mathcal{X}$). *When $G_{x_0}$ is a normal subgroup, we define multiplication on $\mathcal{X}$: $\mathcal{X} \times \mathcal{X} \mapsto \mathcal{X}$ to be $x_1 x_2 := \iota_0(g_1 g_2 G_{x_0})$ for $x_1 = g_1 x_0$ and $x_2 = g_2 x_0$. Under this definition, $x_0$ is the identity element.*

**Lemma 11.** *If $g \in \iota_0^{-1}(x)$, then for any $x' \in \mathcal{X}$, $gx' = xx'$.*

This means that in terms of group action, the group element $g$ is indistinguishable to $x$ on $\mathcal{X}$.

## F.2  GENERAL GROUP ACTION

In this case, $R_g$ can be decomposed into a direct sum of smaller matrices, and all our analysis applies to each of these small matrices.

In the main text, to simplify the notation, we assume that the group action is transitive, i.e., for any $y, y' \in Y$, there exists $g \in G$ so that $gy = y'$. In the following we will show that for general group actions, the conclusion still follows.

*Group orbit*. For any $x \in \mathcal{X}$, Let $G \cdot y := \{gy | g \in G\} \subseteq Y$ be its *orbit*.

**Lemma 12.** *For $y, y' \in G$, either $G \cdot y = G \cdot y'$ (two orbits collapse) or $G \cdot y \cap G \cdot y' \neq \emptyset$ (two orbits are disjoint). Therefore, orbits form a partition of $\mathcal{X}$.*

Let $X/G := \{G \cdot y | x \in \mathcal{X}\}$ be the collection of all orbits. The following lemma tells that the matrix representation $R_g$ can be decomposed into a direct sum (i.e., block diagonal matrix) on each orbit.

**Lemma 13** (Direct sum decomposition of $R_g$).

$$R_g = \bigoplus_{Y' \in Y/G} R_g^{Y'} \tag{115}$$

*and each $R_g^{Y'} \in \mathbb{R}^{|Y'| \times |Y'|}$ is a permutation matrix with $\sum_g R_g^{Y'} = \frac{|G|}{|Y'|} \mathbf{1}\mathbf{1}^\top$.*

*Proof.* By the definition of group orbits, the group action $g$ is closed within each $Y'$. Therefore, $R_g$ is a direct sum (i.e., block-diagonal).

For each element $x \in \mathcal{X}'$, let's check its destination under $G$. It is clear that if two group elements $g, h \in G$ maps $\mathcal{X}$ to the same destination, then

$$gy = hy \iff y = g^{-1}hy \iff g^{-1}h \in G_y \iff h = gG_y \tag{116}$$

where $G_y$ is the stabilizer of $\mathcal{X}$, a subgroup of $G$. Therefore, $g$ and $h$ map $\mathcal{X}$ to the same destination, if and only if they are from the same coset of $G_y$. Therefore, each entry of $\sum_g R_g^{Y'}$ on the column $\mathcal{X}$ equals to the size of cosets of $G_y$, which is $|G_y|$. Furthermore, for $y_1, y_2 \in Y'$, since they belong to the same orbit, there exists $g$ so that $gy_1 = y_2$ and thus for any $g' \in G_{y_1}$, we have

$$g'y_1 = y_1 \iff gg'y_1 = gy_1 = y_2 \iff gg'g^{-1}y_2 = y_2 \iff gg'g^{-1} \in G_{y_2} \tag{117}$$

So there exists bijection between $G_{y_1}$ and $G_{y_2}$. This means that $|G_y|$ is constant for any $x \in \mathcal{X}'$ and thus all elements in $\sum_g R_g^{Y'}$ are equal to $|G|/|Y'|$ (i.e., the number of the group elements that send $\mathcal{X}$ out to various destinations in $Y'$, divided by the possible distinct destinations $|Y'|$, results in the number of times each destination gets hit).  $\square$

## G    PROOFS FOR THE CONTENT IN APPENDIX

**Lemma 7** (Structure of $R_g$). *For any $g \in G$, $R_g$ is a permutation matrix.*

*Proof.* Since every element needs to have a destination, every column of $R_g$ sums to 1, i.e., $\mathbf{1}^\top R_g = \mathbf{1}^\top$. Then we prove that the mapping $y \mapsto gy$ is a bijection. Suppose there exists $y_1, y_2$ so that $gy_1 = gy_2$. Therefore by compatibility we have:

$$g^{-1}(gy_1) = g^{-1}(gy_2) \iff (g^{-1}g)y_1 = (g^{-1}g)y_2 \iff ey_1 = ey_2 \iff y_1 = y_2 \tag{118}$$

So any $g$ is a bijective mapping on $\mathcal{X}$. Since every element of $R_g$ is either 0 or 1, $R_g$ is a permutation matrix.  $\square$

**Lemma 8** (Summation of $R_g$). *If the group action is transitive, then $\sum_{g \in G} R_g = \frac{|G|}{d}\mathbf{1}\mathbf{1}^\top$.*

*Proof.* Simply apply Lemma 13 and notice that for transitive group action, $X/G = \{Y\}$.  $\square$

**Lemma 9** (Co-set Mapping $\iota_0$). *There is a bijection between $\{\iota_0^{-1}(x)\}_{x \in \mathcal{X}}$ and co-sets $[G : G_{x_0}]$ of group stabilizer $G_{x_0} := \{g \in G | gx_0 = x_0\}$, which is a subgroup of $G$ fixing $x_0$.*

*Proof.* First we have

$$\iota_0(g) = \iota_0(h) \iff gy_0 = hy_0 \iff y_0 = g^{-1}hy_0 \iff g^{-1}h \in G_{y_0} \iff h \in gG_{y_0} \tag{119}$$

So for any $y = gy_0$, all elements in $\iota_0^{-1}(y)$ are also in $gG_{y_0}$ and vice versa. The bijection is:

$$\iota_0^{-1}(y) \leftrightarrow gG_{y_0}, \qquad \text{for } y = gy_0 \tag{120}$$

or equivalently,

$$y \leftrightarrow \iota_0(gG_{y_0}) \tag{121}$$

$\square$

**Lemma 11.** *If $g \in \iota_0^{-1}(x)$, then for any $x' \in \mathcal{X}$, $gx' = xx'$.*

*Proof.* For $g \in \iota_0^{-1}(x)$, we have $gx_0 = x$. For any $x' = hx_0$, we have:

$$gx' = ghx_0 = (gh)x_0 \tag{122}$$

On the other hand, by definition, $xx' := \iota_0(ghG_{x_0}) = (gh)x_0$. So for any $x'$, $gx' = xx'$.  $\square$

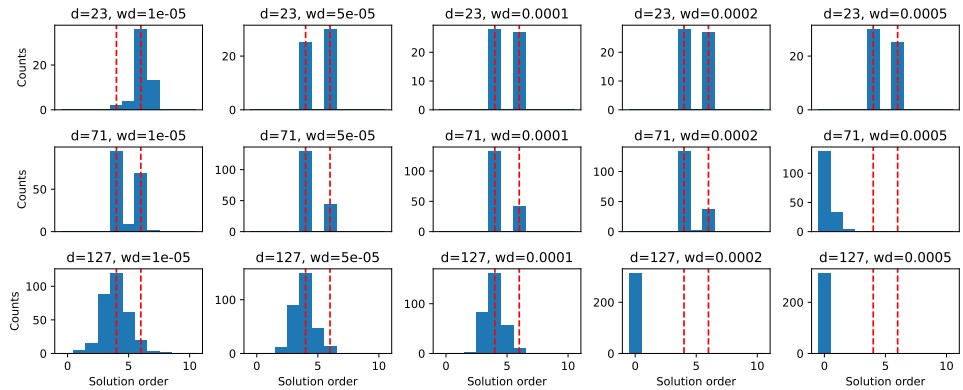

Figure 9: Distribution of solutions with hidden size $q = 256$.

# H ADDITIONAL EXPERIMENTS

**Algorithm to extract factorization from gradient descent solutions**. Given the solutions obtained by gradient descent using Adam optimizer, we first compute the corresponding $z$ via the Fourier transform (that is, Eqn. 12). Here $z = [z_{pkj}]$ is a 3-by-$d$-by-$q$ tensor. Here $d = |G|$ and $q$ is the number of hidden nodes in the 2-layer neural networks.

Then for each frequency $k$, we extract the salient components of $z$ by thresholding with a universal threshold (e.g. $0.05$). The number of salient components (e.g., 6 or 4) is the order of the per-frequency solution.

Suppose we now get $z^{(k)}$ for frequency $k$, which is a 3-by-6 (and thus an order-6) solution. Then we enumerate all possible permutation of 6 hidden nodes ($6! = 720$ possibilities) to find one permutation $\tau$ so that $\|z_{pk\tau(\cdot)} - z_{pk\cdot}^{(1)} \otimes z_{pk\cdot}^{(2)}\|$ is minimized, following ring multiplication defined in Def. 5. Note that for each permutation, we also need to consider whether $\tilde{1} := [-1, -1, 1]$ can be applied to each hidden node $j$ ($\tilde{1}$ is also defined in Tbl. 1). This is because both $z_1 + z_2$ and $z_1 + \tilde{1} * z_2$ have exactly the same values on all monomial potentials (MPs) we consider, due to the fact that $r(\tilde{1}) = 1$ for any $r \in R_g \cup R_c \cup R_n \cup R_*$. Therefore we call $\tilde{1}$ "pseudo-1".

For search efficiency, we therefore first consider the permutation $\tau$ so that $\|z_{ck\tau(\cdot)} - z_{ck\cdot}^{(1)} \otimes z_{ck\cdot}^{(2)}\|$ is minimized, since the component $c$ is invariant to the pseudo-1 transformation $\tilde{1}$, and then for those eligible $\tau$, we search whether $\tilde{1}$ should be applied when considering $p \in \{a, b\}$.

Once we find such $z_1$ and $z_2$, we convert them into their canonical forms $\tilde{z}_1$ and $\tilde{z}_2$ (Def. 8) to eliminate any possible multiplicative term $y$ so that $z_1 = y * \tilde{z}_1$. We then compare the canonical forms (up to complex conjugate) with various order-3 and order-2 partial solutions constructed by CoGO, as detailed in Sec. 5. If their distance is below a certain threshold (e.g., $< 10\%$ of the norm after normalizing both $\hat{z}_1$ and $\hat{z}_2$), then a match is detected.

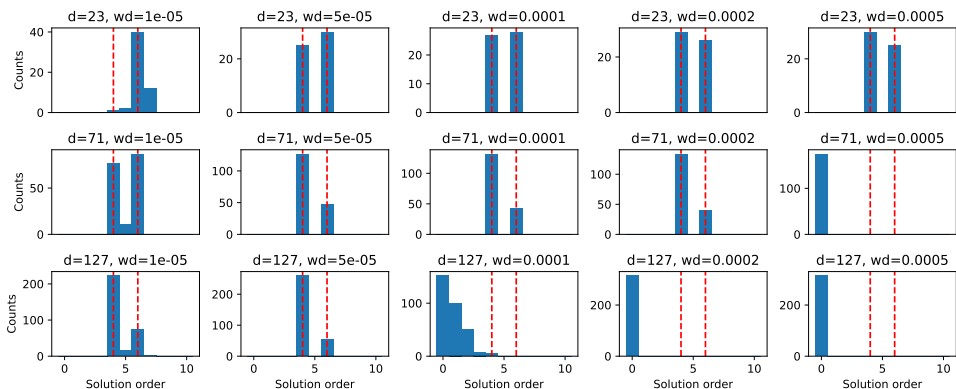

Figure 10: Distribution of solutions with hidden size $q = 512$.

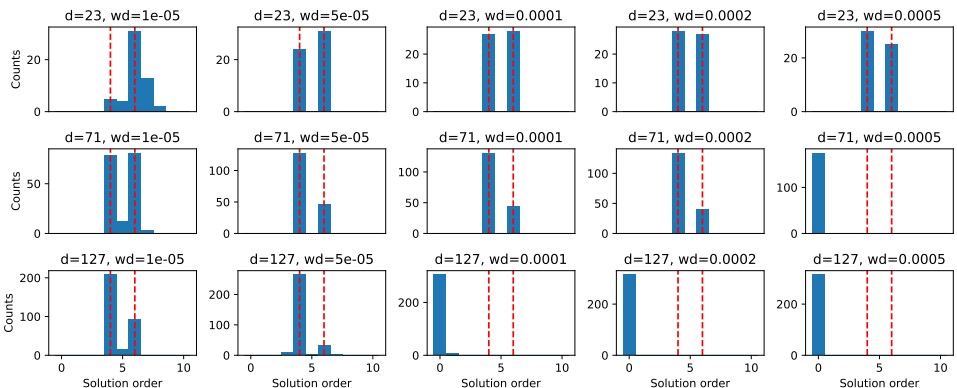

Figure 11: Distribution of solutions with hidden size $q = 1024$.

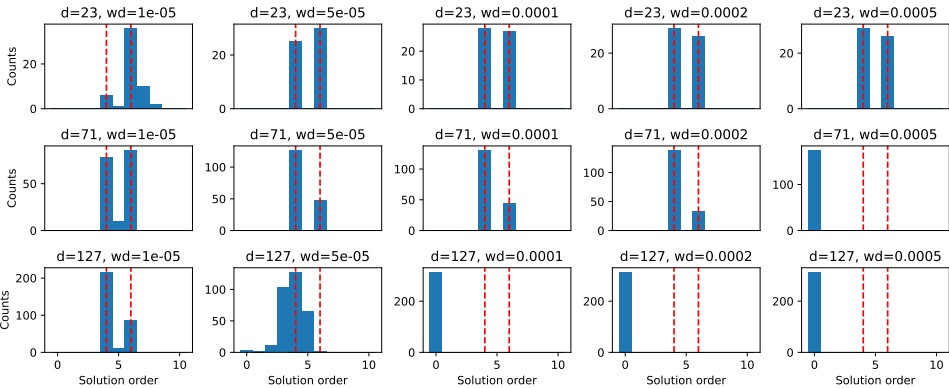

Figure 12: Distribution of solutions with hidden size $q = 2048$.

