# OpenReview forum: "Composing Global Optimizers to Reasoning Tasks via Algebraic Objects in Neural Nets"
_ICLR.cc/2025/Conference — Submitted to ICLR 2025_

### Official Review · Reviewer_orPN · 2024-10-29

**Soundness:** 3
**Presentation:** 2
**Contribution:** 4
**Rating:** 6
**Confidence:** 2

**Summary:**

The work analyzes the 2-layer network training dynamics when learning Abelian group multiplication. Gradient descent matches an analytical solution for optimality.

**Strengths:**

Studies a simple and interesting class of neural networks.
Proves many nice properties of a new mathematical space.
There is probably a nice interpretation of the construction of the solutions in Section 5.2 (but a weakness is that I don't see this expressed in a simple way). Interesting results about behavior of gradient descent in Section 6.

**Weaknesses:**

Numerous grammatical errors ("which are ring homomorphism", "goes to infinite", "is called semi-ring"...)
On the whole, the presentation of technical results is not clear enough to get a good picture of what is happening mathematically.

**Questions:**

What is l[i] in (1)?
Is it important in Section 4.1 that you are looking at solutions in a weight space, or can they just be any fixing of parameters?
Doesn't the loss function itself change when you change the shapes of the parameters?

Clarify the relationship between Input and Output paragraph with what follows.
Be consistent with subscripts with commas or multindices. I'm confused now if they have different meanings.
Clarify the construction alluded to at the beginning of 5.1.
The relationship between weights, w, z, and r should be better clarified. This seems to me like a lot of notation and I don't have the intuition to understand the claims.
Please also explain the essence of the constructions of solutions in 5.2. What is really "going on"?

---

> ### Author Response · Authors · 2024-11-14
> **Rebuttal**
>
> Thanks the reviewer for the insightful comments! We really appreciated it!
>
> We apologize for the grammatical issues and will fix them in the next version. Please check the common rebuttal for the common issues (practicability, connection with grokking, notation, etc). Here are the answers to the specific questions raised by the reviewer.
>
> **Doesn't the loss function itself change when you change the shapes of the parameters?**
>
> The nice structure of Theorem 1 is that it holds regardless of the number of hidden nodes. This is because the monomial potentials (MPs), $r_{k_1k_2k}$ and $r_{pk_1k_2k}$ are a summation over all hidden nodes, and thus is well-defined across different number of hidden nodes. As a result, the expression of Theorem 1 is valid over the entire weight space $\mathcal{Z}$ that contains different weight shapes (the input dimension is $2d$ and the output dimension is $d$, which does not change). This paves the way for our theoretical analysis over the space $\mathcal{Z}$ of all weights of different hidden sizes.
>
> **How the global optimizers are constructed**
>
> The essence here is that we can first construct “partial solutions” to the MSE objectives and then combine them together using algebraic operations (i.e. ring addition and multiplications defined in Def. 5) to form “globally optimal” solutions (i.e. global optimizers). Fig. 1 summarizes the idea.
>
> To define “partial solutions”, we first define a sufficient condition on the weights that leads to global solutions (Lemma 1), which consists of multiple constraints (e.g. certain terms need to be 0, while other terms need to be 1, see Eqn. 4). Then a partial solution is naturally defined as a solution that satisfies only a subset of such constraints.
>
> **Clarify the construction alluded to at the beginning of 5.1**
>
> Intuitively, partial solutions are easier to construct since fewer constraints need to be satisfied. In Sec. 5.1, we construct such solutions via polynomials. The intuition is simple: starting from a order-1 solution $\mathbf{u}$ (order-1 solution means solution with only 1 hidden node) which may not satisfy any constraints, construct a polynomial called $\boldsymbol{\rho(\mathbf{u})}$, so that it will satisfy a subset of such constraints. Then $\boldsymbol{\rho(\mathbf{u})}$ is a partial solution (Table 1 shows a few examples of such partial solutions).
>
> Note that the polynomial construction is not hard: just to consider a fully factored polynomial $\boldsymbol{\rho(\mathbf{u})} = \prod_k (\mathbf{u} - r_k(\mathbf{u}))$, where $r_k = 0$ are the constraints we want. Then for all $k$, $r_k(\boldsymbol{\rho(\mathbf{u})}) = 0$ so $\boldsymbol{\rho(\mathbf{u})}$ satisfies multiple such constraints. Theorem 4 is a more formal version of such arguments.
>
> **Please also explain the essence of the constructions of solutions in 5.2. What is really "going on"?**
>
> The polynomials $\mathbf{z} = \boldsymbol{\rho(\mathbf{u})}$ generated from a single $\mathbf{u}$ are still partial solutions, not global ones. To make global ones, we consider a solution constructed by ring multiplication $\mathbf{z} = \mathbf{z}_1 * \mathbf{z}_2$. Here is the key property we leverage for construction: if partial solution $\mathbf{z}_1$ satisfies $r_a(\mathbf{z}_1) = 0$, and partial solution $\mathbf{z}_2$ satisfies $r_b(\mathbf{z}_2) = 0$, then $\mathbf{z}$ satisfies both $r_a=r_b=0$, since $r_a(\mathbf{z}) = r_a(\mathbf{z}_1*\mathbf{z}_2) = r_a(\mathbf{z}_1)r_a(\mathbf{z}_2) = 0$ (and similar for $r_b$) thanks to the property of ring homomorphism $r_a$ and $r_b$. Lemma 2 is a fancy way of saying it.
>
> Using such a vehicle, in Sec. 5.2, we can construct solutions towards global optimality for each frequency. Both order-6 and order-4 solutions are constructed this way. Summing over frequency yields the final global optimizers (Corollary 2,3,4).

---

> ### Author Response · Authors · 2024-11-22
> **Follow-up**
>
> We hope that the rebuttal could address your concerns. Let us know if you have any further questions. Thanks!

---

> > ### Comment · Reviewer_orPN · 2024-11-27
> > **Response to the new version**
> >
> > Hello, thank you for your revision and additional clarification. I am increasing my score because the presentation changes make the ideas clear enough to understand the main contributions.

---

> > > ### Author Response · Authors · 2024-11-28
> > > **Thanks!**
> > >
> > > We really appreciate your response and score updates! In case you have any additional questions and concerns, let us know.

---

### Official Review · Reviewer_5nTf · 2024-11-02

**Soundness:** 3
**Presentation:** 3
**Contribution:** 3
**Rating:** 5
**Confidence:** 2

**Summary:**

This work considered 2-layer neural networks with quadratic activation and L2 loss on learning group multiplication (an extension of modular addition). It showed that global optimizers can be constructed algebraically from small partial solutions that are optimal only for parts of the loss, due to (1) a semi-ring structure over the weights space and (2) L2 loss being a function of monomial potentials allowing composition of partial solutions into global ones. (2) is shown by representing the network weights and then the loss function using Fourier bases.

It then proposed a systematic approach using the above algebraic structure to construct global optimizers. It used this theoretical framework named CoGO to construct two distinct types of Fourier-based global optimizers of per-frequency order 4 and 6, and a global optimizer of order that correspond to perfect memorization. It empirically showed that most solutions via gradient descent match such constructions. It also analyzed the gradient dynamics, showing that it favors simpler solutions under weight decay, and that overparameterization asymptotically decouples the dynamics.

**Strengths:**

- The work provided a new angle on analyzing the global optimizers for the considered algebraic problem. It analyzed algebraic properties of the weight space and the loss, and then gave sufficient conditions for the global optimizers.
- The study is quite solid and thorough. It provided detailed characterization of the sufficient condition, and also gave a systematic approach to construct global optimizers.

**Weaknesses:**

- The theoretical setup is quite specific: quadratic activation and learning group multiplication. While the analysis is interesting, it is unclear if the results can provide insights into more general settings, in particular those more related to practical scenarios. The work can be strengthened if it can provide some empirical study on more realistic datasets verifying the insights (ie composition structure of the solutions), or provide generalization to more general settings (at least discussion about potential generalization and why).
- The global optimizers constructed by CoGO is only a subset of all possible global optimizers, so the approach only partially characterizes the problem solutions. This weakens the contribution a bit, though the work does provide empirical evidence that most practically obtained solutions are in their construction.
- The presentation can be improved. See several comments below.

**Questions:**

- Line 140: Should mention l[i] is the embedding of the true label for the i-th data point.
- Line 145: I guess l[i] should be the in d-dimension, ie, the embedding of the element g_1[i] g_2[i], rather than the element itself.
- Line 145: How is g_1[i] g_2[i] embedded into l[i]? g_1[i] is using U_{G_1} and g_2[i] is using U_{G_2}, while it's unclear how l[i] is obtained.
- Experiment: how to generate the training data (ie how g_1[i] and g_2[i] are sampled)? The data distribution can significantly impact the solution reached by training, so it needs to be specified for interpreting the empirical result that most solutions reached in experiments match the theoretical construction.

---

> ### Author Response · Authors · 2024-11-14
> **Rebuttal**
>
> Thanks the reviewer for the insightful comments! We really appreciated it!
>
> Please check the common rebuttal for the common issues (practicability, connection with grokking, notation, etc). Here are the answers to the specific questions.
>
> **The approach only partially characterizes the problem solutions**
>
> We totally agree with the reviewers’ assessment. There exist some partial/global solutions that may not be constructed (or factorized) by the current framework. Since the weight space follows a (semi-)ring structure, similar to the integer/polynomial ring, such solutions can be regarded as “prime number” or “prime polynomial”. Characterizing all prime numbers in number theory has been a highly nontrivial open math problem for centuries and we just want to set the proper expectation here for the reviewers. We are also not aware of any existing works doing that for nonlinear neural networks. CoGO at least discovers, sets up the underlying algebraic structures and shows that the practical solutions fall into the algebraic construction, which from our point of view, is already nontrivial.
>
> Despite the challenges, we indeed can prove certain necessary conditions (e.g. for each frequency, the solutions need to be at least order-3 in order to satisfy the first two sets of constraints specified in Lemma 1), and will update the draft later.
>
> **How to generate the training data (ie how g_1[i] and g_2[i] are sampled)?**
>
> We totally agree with the reviewers’ concern that the data distribution can significantly impact the solution reached by training,
>
> For our theoretical study, we assume that $(g_1, g_2)$ follows a uniform distribution, in order to derive Theorem 1 that decomposes MSE loss into several terms of monomial potentials. If $(g_1, g_2)$ are not uniformly distributed, then there will be additional terms in the loss decomposition (Theorem 1)
>
> For our experiments, the training data are constructed as follows. There are $d^2$ distinct pairs of $(g_1, g_2)$, since both $g_1$ and $g_2$ can take values from $0$ to $d-1$ (here $d$ is the $\mathrm{mod}\ d$ in modular addition). The training set is constructed by randomly sampling 90% of $(g_1, g_2)$ pairs out of $d^2$ distinct inputs, and the test set is the remaining 10%. With different random seeds, the training/test partitions are also different. We make sure the empirical experiments match closely with the theoretical setting. Using 95% or even 100% training data gives similar outcomes, and the 10% test set is to verify that the training is on track. Otherwise there will be no validation accuracy and/or the validation accuracy is not accurate due to insufficient data points.
>
> We indeed perceive that if we only use a small portion of the $d^2$ distinct pairs (e.g. using $O(d)$ samples), then only memorization is perceived but no grokking and follow-up generalization, which is consistent with many previous grokking studies. We leave a systematic theoretical study of such topics in future work.

---

> ### Author Response · Authors · 2024-11-22
> **Follow-up**
>
> We hope that the rebuttal could address your concerns. Let us know if you have any further questions. Thanks!

---

> > ### Comment · Reviewer_5nTf · 2024-11-28
> >
> > Thank you for the clarification. However, in general, the response does not address my concern about the scope and significance of the work. So I'll keep my score.

---

> > > ### Author Response · Authors · 2024-11-28
> > > **Thanks for your reply!**
> > >
> > > We are sorry to hear that our rebuttal does not address your concerns. Would you mind telling us which specific concerns have not been addressed yet? In the rebuttal, we provided
> > >
> > > 1. one case that generalizes to a more realistic setting (group action prediction, related to reinforcement learning) in Appendix E;
> > >
> > > 2. several necessary conditions for Lemma 1 (condition of global solutions), see newly added Lemma 5 and 6 that shows no order-2 solutions can satisfy both $R_\mathrm{c}$ and $R_\mathrm{g}$ simultaneously in Lemma 1, and specific structures that order-3 solution must follow;
> > >
> > > 3. an updated version of the draft to improve presentation (in particular for problem settings), as acknowledged by reviewer **orPN**.
> > >
> > > 4. more information about the data distribution and a tentative explanation of how grokking works using our framework.
> > >
> > > Please let us know if you have any specific concerns. Thanks!

---

> > > > ### Author Response · Authors · 2024-12-03
> > > > **Follow-up**
> > > >
> > > > Dear reviewer 5nTf,
> > > >
> > > > The discussion period is about to end. Let us know if you have any specific concerns so that we can address them in time. Thanks.

---

### Official Review · Reviewer_RUmT · 2024-11-03

**Soundness:** 3
**Presentation:** 2
**Contribution:** 3
**Rating:** 6
**Confidence:** 2

**Summary:**

This paper introduces CoGO (Composing Global Optimizers), a theoretical framework for analyzing how 2-layer neural networks learn group operations with quadratic activation and L2 loss. The key insight is discovering a semi-ring algebraic structure in the solution space that allows the construction of global optimizers by composing partial solutions. The authors prove that the weight space has a semi-ring structure and that the loss function consists of monomial potentials with ring homomorphism properties. They also analyze training dynamics to explain why networks prefer simpler Fourier-based solutions over perfect memorization. The theoretical predictions align well with empirical results, showing that about 95% of gradient descent solutions match their constructed solutions.

**Strengths:**

- The work provides theoretical insights into neural network learning mechanisms for group operations. The discovery of algebraic structures (semi-ring) in the weight space and monomial potentials in the loss function offers a fresh perspective on how networks learn structured tasks.
- There's strong empirical validation of the theoretical results. As shown in Table 2, around 95% of gradient descent solutions exactly match their theoretical constructions, with very small factorization errors. This provides concrete evidence that the theoretical framework accurately captures the learning behavior.
- The analysis of training dynamics (Theorem 5 and 6) provides insights into why networks prefer low-order Fourier solutions over perfect memorization. The paper shows that gradient descent with weight decay naturally favors simpler solutions due to topological connectivity between different-order solutions, which is an interesting finding.

**Weaknesses:**

- My major concern is that the loss decomposition approach (Theorem 1) seems limited to scenarios where we already understand the underlying group structure of the data. The paper doesn't address how this framework might generalize to real-world scenarios where the data's algebraic structure is unknown or unclear. This limits the practical applicability of the theoretical insights, e.g., can we decompose the next token prediction loss easily?
- While the training dynamics analysis (particularly around Fourier feature learning and Theorem 5) is interesting, [1] also introduced that the NN prefers to learn Fourier features by gradient descent. Can the author give a more detailed comparison of connections and differences to [1]? The paper could better contextualize its findings with existing work by providing a more detailed comparison of the mechanisms and insights, which would strengthen the paper's contribution.
- The paper mentions connections to grokking in the Conclusion but doesn't fully explore this direction. It would be good to discuss more, e.g., why there is a gap between train loss and test loss in the beginning under the paper’s analysis framework. Given that grokking is a significant phenomenon in neural network learning, especially for arithmetic tasks, a more detailed discussion of how CoGO might explain or relate to grokking would enhance the paper's impact.

[1] Depen Morwani, Benjamin L Edelman, Costin-Andrei Oncescu, Rosie Zhao, and Sham Kakade. Feature emergence via margin maximization: case studies in algebraic tasks. ICLR 2024.

**Questions:**

See the Weaknesses above.

---

> ### Author Response · Authors · 2024-11-14
> **Rebuttal**
>
> Thanks the reviewer for the positive feedbacks! We really appreciated it.
>
> Please check the common rebuttal for the common issues (practicability, connection with grokking, notations, etc)
>
> **A more detailed comparison of connections and differences to [1]?**
>
> The comparison is mentioned in the related work section (line 089-091). We will elaborate here.
>
> **Similarity**: the input setting of [1] and CoGO are exactly the same: two one-hot vectors encoding $g_1$ and $g_2$ are concatenated into a $2d$ vector, which is sent to the 2-layer networks with quadratic activations.
>
> 1. One of the main contributions of CoGO is to discover algebraic structures (the semi-ring structure) in the weight space across different numbers of hidden nodes, and show that the terms in the MSE loss are ring homomorphisms. [1] does not discover such a structure. Thanks to the algebraic structure, the analysis becomes much simpler and CoGO can treat weights of different number of hidden nodes with the same form of loss function.
>
> 2. [1] uses the max-margin framework with a special regularization ($L_{2,3}$ norm), proving that for max-margin solutions, the weight of each neuron needs to be a specific Fourier base of a certain frequency and every frequency is covered by some neurons (Theorem 7 in [1]). Since the framework is max-margin, every neuron needs to contribute to the final margin and thus dead neurons are not allowed. In contrast, CoGO uses MSE (L2) loss, which is arguably more popular, constructs Fourier solutions that are globally optimal to the MSE loss, and also characterizes the fine-grain structures of such solutions (e.g. how many Fourier bases of a certain frequency are needed, their factorization structures) and demonstrates that such constructions matches very well with the gradient descent solutions, at the level of their specific factorization structure.
>
> 3. CoGO also analyzes the topological structures of the global solutions, i.e. global solutions that are connected algebraically via ring multiplication are also topologically connected via a zero-loss curve. [1] does not have such conclusions.
>
> 4. [1] also analyzes Non-abelian groups while CoGO focuses on Abelian groups.
>
> [1] Depen Morwani, Benjamin L Edelman, Costin-Andrei Oncescu, Rosie Zhao, and Sham Kakade. Feature emergence via margin maximization: case studies in algebraic tasks. ICLR 2024.

---

> ### Author Response · Authors · 2024-11-22
> **Follow-up**
>
> We hope that the rebuttal could address your concerns. Let us know if you have any further questions. Thanks!

---

> > ### Comment · Reviewer_RUmT · 2024-11-28
> >
> > I have checked the rebuttal and other reviewers' comments. I appreciate the authors' reply. I would like to keep my positive score.

---

> > > ### Author Response · Authors · 2024-11-28
> > > **Thanks for your reply!**
> > >
> > > We hope our rebuttal addresses your concerns. Let us know if you have more questions!

---

### Author Response · Authors · 2024-11-14
**Rebuttal**

We appreciate all reviewers for their insightful comments. All reviewers agree that our work studies the algebraic structures of solutions to an interesting class of nonlinear networks in a novel way, and demonstrates that our theoretical construction matches well with the solutions obtained by gradient descent, with relatively thorough experiments. Besides, we also provide analysis of gradient dynamics and show how simple solutions are encouraged by gradient descent, which are interesting to reviewers [**orPN**, **RUmT**].

We apologize for any presentation issues and will fix them (e.g. grammatical issues) in the next revision, which will be uploaded later in the discussion period. To clarify the confusion as soon as possible, we answer the major concerns of the reviewers as follows:

## Practical use case of the theoretical framework and generalized to real-world scenarios [**RUmT**, **5nTf**]

Here we provide several use cases and possible extension of CoGO. We want to emphasize that it remains a highly nontrivial problem to derive a consistent theoretical framework that clearly tells the solution structures of neural networks, and at the same time, directly connects with the practical scenarios.

**Real-world scenarios where the underlying algebraic structure is unknown?** [**RUmT**, **5nTf**]

First we want to clarify that when training neural networks for group multiplication tasks (like modular addition), the neural network doesn’t know its algebraic structure and treat it as a regular classification/regression problem. The hidden algebraic structure of the task leads to the emergent structure of the global solutions.

While in the main text, we focus on group multiplication (and modular addition) to simplify notation and make the main story more clear, CoGO can be applied to more general cases, such as group action prediction problems, as shown in Appendix E.  In such problems, the input/output mapping $(g_1, g_2) \mapsto g_1 g_2$ is now generalized to $(g, x) \mapsto gx$, where $g$ is an action, $x \in \mathcal{X}$ is some state and $x’ = gx$ is an altered state after the action $g$ is applied to $x$.

This setting corresponds to real-world scenarios. For example, in reinforcement learning, we want to model how the world state changes $x \mapsto x’$ after an action $g$ is applied.

In Appendix E, we are able to show that if (1) all actions form a group $G$ and (2) the operation $x' = gx$ satisfies two properties (identity and compatibility, see line 1263-1267), then if $G$ is Abelian, then the set $\mathcal{X}$ can be decomposed into a union of disjoint $\mathcal{X} = \bigcup_l \mathcal{X}_l$, in which each transitive component $\mathcal{X}_l$ is *isomorphic* to a subgroup of $G$. Then in each $\mathcal{X}_l$, we could define its own subgroup multiplication operations, the action of $g$ is also restricted to this subgroup, and our CoGO analysis in the main text can still be applied. Note that again no information about the structure of $\mathcal{X}$ is known by the training algorithm.

**Explain grokking with this framework** [**RUmT**]

Our analysis gives some intuition regarding grokking. Theorem 1 shows that the loss function is a summation of a linear term ($r_{kkk}$) and a few quadratic (sum of squares) terms in the Fourier domain. When the weights are small, the quadratic terms are much smaller than the linear term and the weights grow at a uniform pace. This means that all the weights are similar in magnitude (in the Fourier domain) and memorization happens (check the perfect memorization solution in Eqn. 8 in Corollary 4, in which all weights in Fourier domains have the same magnitude). However, when the weight magnitude becomes larger, the quadratic terms (as well as weight decay) catch up, which leads to specialization of hidden neurons into different frequencies, as shown in Fig. 6, which is the generalization solutions (order-4 and order-6 solutions in Corollary 2 and 3).

From this analysis, it is clear that we need a small learning rate to demonstrate the entire phase transition process, and a fairly large weight decay to trigger node specialization, converging to low-order solutions, as suggested in Theorem 6. This simple analysis seems to align with existing studies [2], i.e. small learning rate and reasonably large weight decay lead to grokking, and the model stays in memorization with super small weight decay (e.g., Fig. 7(b) and 8(b) in [2]).

Note that this is a very rough qualitative analysis and lots of questions remain, e.g. percentage of training samples out of all possible $d^2$ distinct pairs to enable such transition, etc. The current framework will lead to additional terms in Theorem 1, if the training distribution is no longer uniform across input pair $(g_1, g_2)$. This makes analysis complicated. Therefore, we leave it for future work.

[2] Towards Understanding Grokking: An Effective Theory of Representation Learning (https://arxiv.org/abs/2205.10343)

---

> ### Author Response · Authors · 2024-11-14
> **Rebuttal #2**
>
> ## Notations
>
> > What is $l[i]$?
>
> $l[i]$ is indeed a $d$-dimensional one-hot vector of ground truth label, it is not the element $g_1[i]g_2[i]$ itself. Thanks **5nTf** for pointing it out and we will revise the paper.
>
> > How is $g_1[i] g_2[i]$ embedded into $l[i]$? $g_1[i]$ is using $U_{G_1}$ and $g_2[i]$ is using $U_{G_2}$?
>
> $l[i]$ is a $d$-dimensional one-hot vector of label $g_1[i]g_2[i]$. $U_{G_1}$ and $U_{G_2}$ are column-orthonormal matrices and their column subspaces are also orthogonal to each other. One simple example is that $g_1[i]$ and $g_2[i]$ are encoded in $d$-dimensional one-hot vectors respectively, and the two one-hot vectors are concatenated into a $2d$ vector as the input of the 2-layer neural network. Here $d = |G|$ is the size of the group $G$ and also the $\mathrm{mod}\ d$ of the modular addition.

---

### Author Response · Authors · 2024-11-22
**Update on the draft**

We have updated a new revision of the paper. It addresses:

1. Notation issues pointed by the reviewers **5nTf** and **orPN**. Now the problem setting should be much easier to understand. The changes are highlighted in blue.
2. Fixing grammatical errors pointed by the reviewers.
3. Added Lemma 5 and 6 in appendix (page 21-23) to characterize necessary conditions of the solutions that satisfy the global optimality condition (Lemma 1). Thus it partially address the concerns raised by reviewer **5nTf**.

Thanks all the reviewers for the helpful comments to make the work better!

---

### Meta-Review · Area_Chair_53R3 · 2024-12-20

**Metareview:**

Dear Authors,

Thank you for your valuable contribution to ICLR and the ML community. Your submitted paper has undergone a rigorous review process, and I have carefully read and considered the feedback provided by the reviewers.

This work proposes considers two-layer neural networks with quadratic activation trained for learning group multiplication using L2 loss. The results show that optimizers can be constructed algebraically from small partial solutions that are optimal only for parts of the loss due to algebraic properties of the weight space.

The paper received borderline review scores (6,6,5). Reviewers pointed out certain issues including (i) narrow and specific nature of the theory, (ii) practical relevance of the quadratic activation networks, (iii) characterizing only a subset of all possible global optimizers. Thank you for providing a detailed rebuttal. However, the rebuttal was not convincing enough for the reviewers to increase their scores from borderline.

Given the current form of the paper and the reviewer discussion, I regret to inform you that I am unable to recommend the acceptance of the paper for publication at ICLR. I want to emphasize that this decision should not be viewed as a discouragement. In fact, the reviewers and I believe that your work has valuable insights, a quite deep and interesting theory and, with further development and refinement, it can make a meaningful impact on the field.

I encourage you to carefully address the feedback provided by the reviewers and consider resubmitting the paper. Please use the comments and suggestions in the reviews to improve and refine your work.

Best,
AC

**Additional Comments On Reviewer Discussion:**

Reviewer 5nTf and others pointed out critical issues including (i) narrow and specific nature of the theory, (ii) practical relevance of the quadratic activation networks, (iii) characterizing only a subset of all possible global optimizers. The authors provided a detailed rebuttal, however, the rebuttal was not convincing enough for the reviewers to increase their scores from borderline.

---

### Decision · Program_Chairs · 2025-01-22

Reject